# Sub-part-per-trillion test of the Standard Model with atomic hydrogen

Lothar Maisenbacher[1,4 ✉], Vitaly Wirthl[1], Arthur Matveev[1], Alexey Grinin[1,5], Randolf Pohl[2], Theodor W. Hänsch[1,3] & Thomas Udem[1,3]

Quantum electrodynamics (QED), the first relativistic quantum field theory, describes light–matter interactions at a fundamental level and is one of the pillars of the Standard Model (SM). Through the extraordinary precision of QED, the SM predicts the energy levels of simple systems such as the hydrogen atom with up to 13 significant digits[1], making hydrogen spectroscopy an ideal test bed. The consistency of physical constants extracted from different transitions in hydrogen using QED, such as the proton charge radius $r_p$, constitutes a test of the theory. However, values of $r_p$ from recent measurements[2–7] of atomic hydrogen are partly discrepant with each other and with a more precise value from spectroscopy of muonic hydrogen[8,9]. This prevents a test of QED at the level of experimental uncertainties. Here we present a measurement of the 2S–6P transition in atomic hydrogen with sufficient precision to distinguish between the discrepant values of $r_p$ and enable rigorous testing of QED and the SM overall. Our result $\nu_{2S–6P} = 730{,}690{,}248{,}610.79(48)$ kHz gives a value of $r_p = 0.8406(15)$ fm at least 2.5-fold more precise than from other atomic hydrogen determinations and in excellent agreement with the muonic value. The SM prediction of the transition frequency (730,690,248,610.79(23) kHz) is in excellent agreement with our result, testing the SM to 0.7 parts per trillion (ppt) and, specifically, bound-state QED corrections to 0.5 parts per million (ppm), their most precise test so far.

The binding energy of atomic hydrogen can be expressed as[1]

$$E_{nlJ} = chR_\infty \left( f_{nlJ}^{\text{Dirac}}\left(\alpha, \frac{m_p}{m_e}\right) + f_{nlJ}^{\text{QED}}\left(\alpha, \frac{m_p}{m_e}, \dots\right) + \delta_{l0} \frac{C_{NS}}{n^3} r_p^2 \right), \quad (1)$$

in which $n$, $l$ and $J$ are, respectively, the principal, orbital and total electronic angular momentum quantum numbers of the energy level of interest. $f_{nlJ}^{\text{Dirac}}$ is the Dirac eigenvalue ($\propto 1/n^2$ in leading order), whereas the second term $f_{nlJ}^{\text{QED}}$ ($\propto 1/n^3$ in leading order) contains the corrections from bound-state QED, such as self-energy and vacuum polarization, including muonic and hadronic contributions[1]. Both terms depend on the fine-structure constant $\alpha$ and the electron-to-proton mass ratio $m_p/m_e$, which are known with sufficient accuracy from other experiments that do not require bound-state QED (refs. 1,10–14). The third term is the leading-order nuclear size correction for S-states ($l = 0$), accounting for the finite root-mean-square (rms) charge radius of the proton, $r_p$. The second and third terms constitute the Lamb shift and contribute about 1 ppm and 100 ppt, respectively, to the 2S–6P transition frequency (Extended Data Table 1). The unitless terms are converted to SI units (International System of Units) using the Rydberg constant $R_\infty$.

To compare measured energy levels or transition frequencies with equation (1), $r_p$ and $R_\infty$ must be known (speed of light in vacuum $c$ and Planck's constant $h$ are defined). In practice, $r_p$ and $R_\infty$ are largely determined from such measurements themselves and more than two measurements of distinct transitions are necessary to test equation (1).

A special case is the determination of $r_p$ with laser spectroscopy of muonic hydrogen[8,9,15]. In this exotic atom, the electron is replaced with a negative muon, whose larger mass increases $C_{NS}$ of equation (1) by four orders of magnitude, allowing a precise determination of $r_p$ without requiring other high-precision input. Because the nuclear size correction scales as $1/n^3$ as the other QED corrections, discrepant $r_p$ values from atomic and muonic hydrogen can indicate missing or incomplete QED terms (Methods). Notably, the value of $r_p$ from the muonic measurement was found to be significantly smaller (>5$\sigma$) than the then-established value (CODATA 2014 (ref. 16)).

This proton radius puzzle led to extensive research efforts[2–7,17]. We first addressed it with a precision measurement of the 2S–4P transition in atomic hydrogen[2], which favoured the muonic result, but could not conclusively (>5$\sigma$) rule out the previous value. Subsequent measurements in atomic hydrogen have followed[3–7] but they are partly discrepant with the muonic value and with each other, and none is precise enough to conclusively test the muonic value, as visualized in Fig. 1. Until now, this has prevented a verification of QED at the level of experimental uncertainties.

Here we report on laser spectroscopy of the 2S–6P transition in atomic hydrogen with sufficient precision to distinguish between the discrepant values of $r_p$. This precision corresponds to finding the transition frequency to one part in 15,000 of the experimental linewidth, to our knowledge unprecedented for laser spectroscopy, requiring a thorough understanding of any asymmetric distortions of the line

[1]Max-Planck-Institut für Quantenoptik, Garching, Germany. [2]Johannes Gutenberg-Universität Mainz, Mainz, Germany. [3]Ludwig-Maximilians-Universität München, Munich, Germany. [4]Present address: University of California, Berkeley, Berkeley, CA, USA. [5]Present address: Northwestern University, Evanston, IL, USA. ✉e-mail: lothar.maisenbacher@mpq.mpg.de

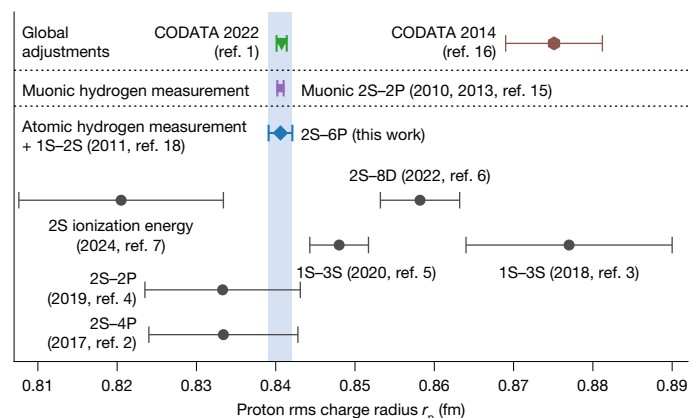

**Fig. 1 | Proton rms charge radius $r_p$.** Previous determinations of $r_p$ from atomic hydrogen spectroscopy (refs. 2–7; black circles) are partly discrepant with each other and with the value of $r_p$ from spectroscopy of muonic hydrogen (refs. 8,9,15; violet square) and therefore could not conclusively resolve the initial $5.6\sigma$ discrepancy between the 2010 muonic value and the then-established larger value (as summarized in the CODATA 2014 global adjustment of fundamental constants[16]; brown hexagon). The value of $r_p$ from atomic hydrogen spectroscopy of the 2S–6P transition in this work (blue bar and diamond) is at least 2.5-fold more precise than other atomic hydrogen determinations and in excellent agreement with the muonic value. It disagrees with the fourfold less precise CODATA 2014 value by $5.5\sigma$. The $r_p$ values are determined by combining each measurement with the 1S–2S transition frequency[18] and equation (1) (Pearson correlation coefficient $r < 0.05$ between $r_p$ values). The most recent CODATA 2022 global adjustment (ref. 1; green triangle) essentially corresponds to the muonic value owing to the exceptionally low uncertainty of the latter. Error bars show one-standard-deviation uncertainties. Electron–proton scattering data are not shown, as different analyses give significantly different values of $r_p$ (refs. 1,48). Lattice QCD calculations of $r_p$ show promise but are not yet competitive[49]. See Extended Data Fig. 2 for the Rydberg constant $R_\infty$ from atomic hydrogen combined with the muonic value of $r_p$.

shape at that level and a large experimental signal-to-noise ratio. By combining our measurement with the precisely known 1S–2S transition frequency[18], we determine $r_p$ with 2.5-fold higher precision than the previous best determination from atomic hydrogen[5] (Fig. 1). Our value of $r_p$ is in excellent agreement with the muonic value but fourfold more precise than and in significant disagreement ($5.5\sigma$) with the CODATA 2014 value[16]. Consequently, we use the muonic value of $r_p$ as input to equation (1) (along with the 1S–2S transition frequency), which allows us to compare the SM prediction of the 2S–6P transition frequency with our measurement. This constitutes a test of the SM to 0.7 ppt and of bound-state QED corrections to 0.5 ppm.

## Principle of the measurement

### 2S–6P transition
We study the 2S–6P transition in a cryogenic beam of hydrogen atoms using Doppler-free one-photon laser spectroscopy. Although the transition has been previously observed with laser spectroscopy[19,20], this work presents a substantial improvement. Using a linearly polarized, 410-nm spectroscopy laser, we alternately examine two dipole-allowed transitions from the metastable initial $2S_{1/2}^{F=0}$, $m_F = 0$ level: the $2S–6P_{1/2}$ transition to the $6P_{1/2}^{F=1}$, $m_F = 0$ level and the $2S–6P_{3/2}$ transition to the $6P_{3/2}^{F=1}$, $m_F = 0$ level, as shown in Fig. 2a ($F$, total angular momentum quantum number; $m_F$, magnetic quantum number). The excited 6P levels rapidly decay, directly or through cascades, to the 1S and 2S manifolds, resulting in a $\Gamma = 3.90$ MHz natural transition linewidth. These decays, predominantly the direct Lyman-$\varepsilon$ decay to the 1S manifold, are the experimental signal and the fluorescence line shape is observed in line scans by recording this signal at different

spectroscopy laser detunings. A fraction of $\gamma_{ei}/\Gamma = 3.9\%$ or $7.9\%$ of decays from the excited level lead back to the initial 2S level for the $2S–6P_{1/2}$ or $2S–6P_{3/2}$ transitions, respectively. Notably, quantum interference (QI) between excitation–decay paths that go through either excited level but lead to the same final level can cause substantial distortions of the fluorescence line shape. The associated line shifts are on the order of $\Gamma^2/\Delta\nu_{FS}(6P) \approx \Gamma/100$ (refs. 2,21,22), in which $\Delta\nu_{FS}(6P) \approx 405$ MHz is the 6P fine-structure splitting between the excited levels. Because the magnitude and sign of the distortions depend on the detection direction (relative to the laser polarization), here we use a large detection solid angle and a magic polarization angle to strongly suppress the shift[22].

### Experimental apparatus
The key in-vacuum components of the experimental apparatus (described in detail in ref. 23) are shown in Fig. 2b. Briefly, a cryogenic beam of hydrogen atoms is formed by a copper nozzle (circular aperture with $r_1 = 1$ mm radius) held at temperature $T_N = 4.8$ K. The atoms are prepared in the initial $2S_{1/2}^{F=0}$, $m_F = 0$ level by Doppler-free two-photon excitation from the 1S ground level with a preparation laser (243 nm wavelength; Fig. 2a,b), collinear with the atomic beam. The divergence of the atomic beam is limited to approximately 10 mrad in the transverse ($x$) direction by a collimating aperture (1.2 mm width, placed 154 mm after the nozzle).

The atomic beam enters the 2S–6P spectroscopy region inside a cylindrical detector assembly, in which, at a distance $L = 204$ mm from the nozzle, it crosses counterpropagating (along $x$) spectroscopy laser beams at an adjustable atomic beam offset angle $|\alpha_0| = 0$–12 mrad from the orthogonal. In the ideal case of laser beams with identical wavefront curvature and power, this excitation scheme produces a line shape whose centre of mass is free of first-order Doppler shifts (but not necessarily free of Doppler broadening), as the interaction with the respective beams results in Doppler shifts of equal magnitude but opposite sign. Here an active fibre-based retroreflector (AFR)[24–26], consisting of polarization-maintaining fibre, collimator and high-reflectivity mirror, generates the required high-quality, wavefront-retracing beams (2.2 mm $1/e^2$ intensity radius; $P_{2S–6P} = 5$–30 μW power in each beam). The AFR is attached to the rotatable detector cylinder, allowing an in situ adjustment of $\alpha_0$ (1 mrad accuracy). We align $\alpha_0$ close to zero to avoid splitting the line shape into two Doppler components from the counterpropagating beams, except when characterizing the light force shift (LFS; see below). The polarization angle $\theta_L$ of the laser beams, relative to the axis of the cylinder (along $y$), is set to $56.5°$, the magic angle at which QI distortions are suppressed, or orthogonal to it ($146.5°$).

The photons emitted by the 6P decay eject photoelectrons from the cylinder walls, which are drawn to and counted with channel electron multipliers at the top and bottom of the cylinder (top and bottom detectors). Each detector covers >20% of the total solid angle, further suppressing QI distortions. A segmented Faraday cage surrounds the spectroscopy region, shielding external electric fields and allowing the application of bias fields to characterize stray electric fields and the dc-Stark shift caused by such fields (Methods).

### Doppler-free one-photon spectroscopy
We investigate different atomic velocity groups $\tau_i$ ($i = 1$–16; Extended Data Table 2) by periodically blocking the 1S–2S preparation laser, thereby intermittently stopping the production of 2S atoms, and recording the signal as a function of delay time $\tau = 10$–2,560 μs. The longer the $\tau$, the slower the 2S atoms contributing to the signal. The speed distribution of atoms contributing to the signal is well described by a Maxwell–Boltzmann flux distribution with an extra multiplicative factor $\exp(-\upsilon_{cut-off}/\upsilon)$ with a characteristic cut-off speed $\upsilon_{cut-off} \approx 50$ m s$^{-1}$, accounting for the loss of slower atoms through collisions (Methods). The mean speeds $\bar{\upsilon}$ of the velocity groups cover a wide range (253(5) to 65(1) m s$^{-1}$; overall mean speed $\langle\bar{\upsilon}\rangle = 195(6)$ m s$^{-1}$)

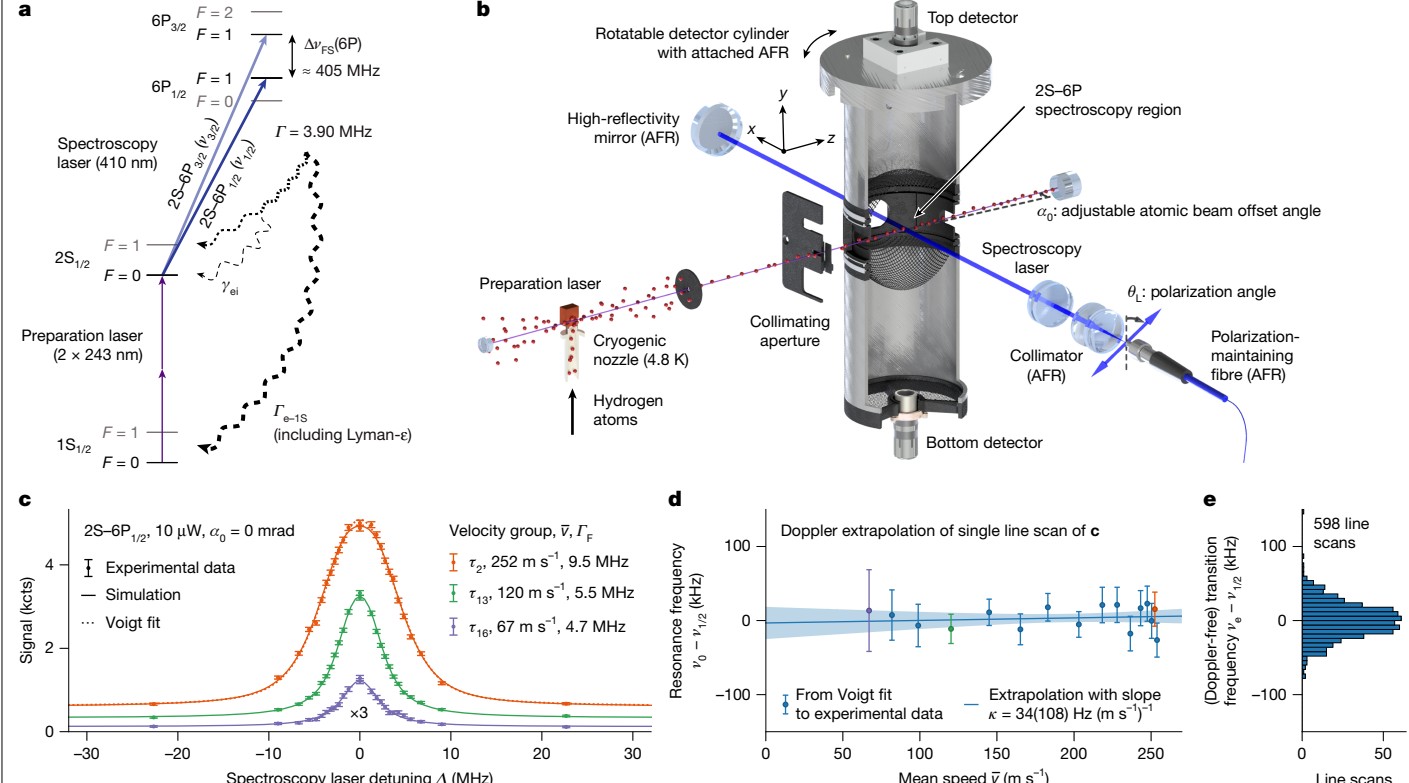

**Fig. 2 | Doppler-free one-photon spectroscopy of the 2S–6P transition.**
**a**, Relevant level scheme of atomic hydrogen (not to scale). Solid arrows indicate laser-driven transitions and dashed arrows indicate spontaneous decay. Spectroscopy laser is on resonance with $2S–6P_{1/2}$ (dark blue) or $2S–6P_{3/2}$ (light blue) transitions. Levels shown in grey are not resonantly coupled by lasers.
**b**, Key components of experimental apparatus (to scale, cutaway). Dashed black lines (orthogonal to spectroscopy laser) visualize atomic beam offset angle $\alpha_0$ (exaggerated) and laser polarization angle $\theta_L$. **c**, Typical line scan of the $2S–6P_{1/2}$ transition (acquired within 40 s; full detuning range is ±50 MHz; Methods). The fluorescence signal (top detector; kcts, kilocounts) is recorded for different velocity groups $\tau_i$ with mean speeds $\bar{v}$, with three shown here (orange, green and purple circles; $\tau_{16}$ scaled (×3) for visibility). Error bars show expected one-standard-deviation ($\sigma$) shot noise. The FWHM linewidth $\Gamma_F$

reduces for slower velocity groups as Doppler broadening reduces. Solid lines show simulated line shape (scaled and offset to match the signal) and the dotted orange line shows the Voigt line shape fit to $\tau_2$ (see Extended Data Fig. 1 for fit residuals). $P_{2S–6P} = 10\ \mu\text{W}$ spectroscopy laser power and $\alpha_0 = 0$ mrad were used.
**d**, Resonance frequencies $\nu_0$ (circles) determined from Voigt line shape fits to velocity groups of scan of panel **c** versus $\bar{v}$. Error bars show $1\sigma$ fit uncertainty. Extrapolation to zero speed (blue line; blue shading, $1\sigma$ confidence interval) gives Doppler-free transition frequency $\nu_e$ and Doppler slope $\kappa$. $\nu_0$ has been corrected for light force, QI and second-order Doppler shifts and all other corrections. **e**, Histogram of all 598 detector-averaged Doppler-free transition frequencies $\nu_e$ (data group G3 of Extended Data Table 3) determined with the same experimental parameters as the line scan of panels **c** and **d**.

and their transverse velocities are approximately Gaussian distributed with a full width at half maximum (FWHM) ranging from 3.4 to 0.6 m s⁻¹ (parentheses give the standard deviation over the data groups of Extended Data Table 3).

Figure 2c shows the top detector signal for three velocity groups (circles) for a typical line scan of the $2S–6P_{1/2}$ transition. Fast velocity groups, such as $\tau_2$ (orange circles), have a FWHM linewidth $\Gamma_F$ substantially larger than the natural linewidth $\Gamma$, owing to Doppler broadening from their large transverse velocity widths. Conversely, for slower velocity groups (green and purple circles), $\Gamma_F$ approaches $\Gamma$ as the transverse velocity width decreases. Overall, $\Gamma_F$ ranges from 9.6(4) to 5.5(6) MHz (Extended Data Table 2).

Our line shape simulations (solid lines; here the QI model is shown; Methods) are in excellent agreement with the experimental data, reproducing the Doppler broadening with $v_{\text{cut-off}}$ as the only free parameter. The resonance frequency $\nu_0$ of each velocity group is determined by fitting simple Voigt or Voigt doublet line shapes (dotted line; Methods) to the data. This differs from the approach taken in our previous 2S–4P measurement[2], in which a theoretically motivated, asymmetric line shape model was used to account for QI distortions. Here the QI distortions are much smaller, owing to the magic polarization angle and large detection solid angle, and distortions from the LFS dominate (Extended Data Fig. 1), for which no equivalent line shape

model is known to us. Instead, we fit the same simple line shape model to the simulations and correct the experimental resonance frequency with our LFS and QI simulations.

To remove any residual first-order Doppler shift, we use the linear model $\nu_0 = \nu_e + \kappa\bar{v}$ to extrapolate the resonance frequencies $\nu_0$ of each line scan to zero speed (Fig. 2d). This gives the Doppler-free transition frequency $\nu_e$, the Doppler slope $\kappa$ and the resulting effective frequency correction $\Delta\nu_e = -\kappa\langle\bar{v}\rangle$ (Methods). Figure 2e shows $\nu_e$ of all 598 line scans recorded with the same experimental parameters as in Fig. 2c,d. In total, the dataset presented here contains 3,155 line scans, acquired in three measurement runs (A, B and C) and grouped into 17 experimental parameter combinations (data groups; Extended Data Table 3).

## Light force shift

Just as light waves diffract on a periodic structure, matter waves can diffract on the periodic structure formed by a standing wave of light, as first predicted for electrons by Kapitza and Dirac[27]. Here a similar diffraction can occur as the atoms, which may be understood as matter waves, cross the standing intensity wave formed by the counter-propagating spectroscopy laser beams used to suppress the first-order Doppler shift (Fig. 2b). This effect, along with scattering on the standing wave, leads to a distortion of the fluorescence line shape and

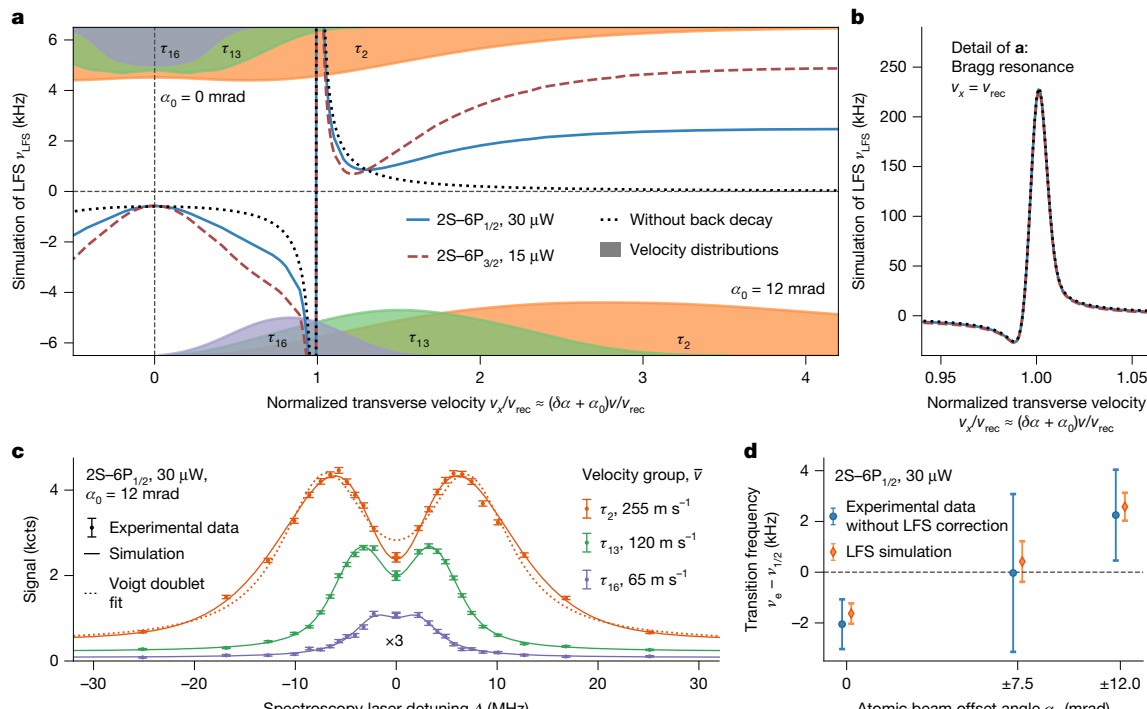

**Fig. 3 | Simulation and measurement of LFS. a**, Simulation of LFS $\nu_{LFS}$, resulting from interaction of atoms with the standing intensity wave (periodicity $\lambda/2 = 205$ nm) formed by counterpropagating spectroscopy laser beams. The simulation describes the atoms as delocalized over nodes and antinodes, crossing the standing wave with speed $\nu = 200$ m s$^{-1}$ and at angle $\delta\alpha + \alpha_0$ to orthogonal (transverse velocity of $\nu_x \approx \nu(\delta\alpha + \alpha_0)$; Fig. 2b). Results are shown for 2S–6P$_{1/2}$ (solid blue line) and 2S–6P$_{3/2}$ (dashed red line) transitions (at equal Rabi frequency), for which, respectively, 3.9% and 7.9% of the 6P level decays lead back to the 2S level (Fig. 2a). The hypothetical situation without back decay is also shown (dotted black line). At $\nu_x = \nu_{rec} \approx 0.97$ m s$^{-1}$ (the recoil velocity), a Bragg resonance occurs, leading to a large shift. The shift is symmetric in $\nu_x$ and

negative (positive) for $|\nu_x| \lesssim \nu_{rec}$ ($|\nu_x| \gtrsim \nu_{rec}$). The simulated distribution (smoothed and scaled for visibility) of $\nu_x$ for an atomic beam offset angle $\alpha_0$ of 0 mrad (12 mrad) is indicated at the top (bottom) for velocity groups $\tau_2$, $\tau_{13}$ and $\tau_{16}$ (arbitrary vertical units). **b**, Detail of the Bragg resonance. **c**, Similar to Fig. 2c but for a line scan with $\alpha_0 = 12$ mrad, which splits the line into two Doppler components. The line is fitted with Voigt doublet line shape (Methods). **d**, Doppler-free transition frequency $\nu_e$ (blue circles) of the 2S–6P$_{1/2}$ transition for data taken at $P_{2S-6P} = 30$ μW and $|\alpha_0| = 0$, 7.5 and 12 mrad and corrected for all systematic effects except LFS. The simulation of LFS (orange circles) is in excellent agreement with the experimental data. Error bars show combined statistical and systematic uncertainty (experimental data) or uncertainty from input parameters (simulation).

a line shift (LFS). Although such shifts have been observed in other experiments using standing waves[28–30], the behaviour and size of the shift and the necessary theoretical treatment are highly dependent on the exact experimental conditions. A travelling wave can be used to avoid the LFS[31] but this approach has not yet demonstrated the level of Doppler shift suppression required here.

Diffraction of matter waves, as for light waves, requires some degree of spatial coherence, with diffraction becoming important when the atoms' transverse coherence length $l_{c,t}$ along the standing wave is comparable with its periodicity of $\lambda/2 = 205$ nm. Treating the cryogenic nozzle as a thermal source of atoms, we find $l_{c,t}$ to be $\lambda_{dB,th} = 0.8$ nm (ref. 32) and therefore much smaller than $\lambda/2$, in which $\lambda_{dB,th} = \sqrt{h^2 k_B T_N/2\pi m_H}$ is the thermal de Broglie wavelength ($m_H$, hydrogen mass; $k_B$, Boltzmann constant). However, as is well known from the van Cittert–Zernike theorem[33], the transverse coherence length is enhanced by propagation, as seen in matter wave interference of large molecules from a thermal source[34]. At a distance $L$ from the nozzle, this results in

$$l_{c,t} \approx (L/r_l)\lambda_{dB}/\pi \qquad (2)$$

for atoms with a speed $\nu$ and de Broglie wavelength of $\lambda_{dB} = h/m_H\nu$. For $\nu = 200$ m s$^{-1}$, the mean speed of atoms examined in the experiment, $l_{c,t}$ is 129 nm at the standing wave and hence comparable with its periodicity. We therefore need to model the atoms as partially coherent matter waves, that is, the atoms are partially delocalized over the standing wave, unlike the localized description[35] valid for different experimental conditions[28,29].

We use the Wigner function[36] to find a quantum-mechanical description of the atomic beam, quantizing the motion along the standing wave while treating other directions classically. This results in a comparable value of $l_{c,t}$ as estimated above[23]. The interaction of the atoms with the standing wave can be described in the combined basis of internal energy levels and external momenta along the standing wave (Supplementary Methods). The absorption of a photon from the forward-travelling (+) or backward-travelling (−) beam excites atoms from the 2S to the 6P level and changes their momentum along the beams by $\pm\hbar K_L$ (wavenumber $K_L = 2\pi/\lambda$; $\hbar = h/2\pi$). Likewise, stimulated emission into either beam returns atoms to the 2S level and changes their momentum by $\mp\hbar K_L$. This corresponds to a change in velocity by the recoil velocity, $\nu_{rec} = \hbar K_L/m_H \approx 0.97$ m s$^{-1}$, comparable with the typical transverse velocity $\nu_x$ of the atoms. Because the 6P level predominantly spontaneously decays to the 1S ground level, from which there is no re-excitation while the signal is recorded, the maximum relevant momentum change is here $\pm4\hbar k$.

The small fraction $\gamma_{ei}/\Gamma$ of the spontaneous decays back to the initial 2S level leads to a random momentum change along the direction of the standing wave, as the direction of the emitted photon is randomly distributed[37]. The associated Doppler shift of such a decay (at most twice the recoil shift of $\Delta\nu_{rec} = 1,176.03$ kHz) is below the natural linewidth $\Gamma$ and therefore the atom can be re-excited to the 6P level and ultimately contribute to the signal. However, because $\gamma_{ei}/\Gamma \ll 1$, only at most one such back decay is relevant here.

We first simulate the interaction of the atoms with the standing wave for an atom initially in a momentum eigenstate, that is, as a fully

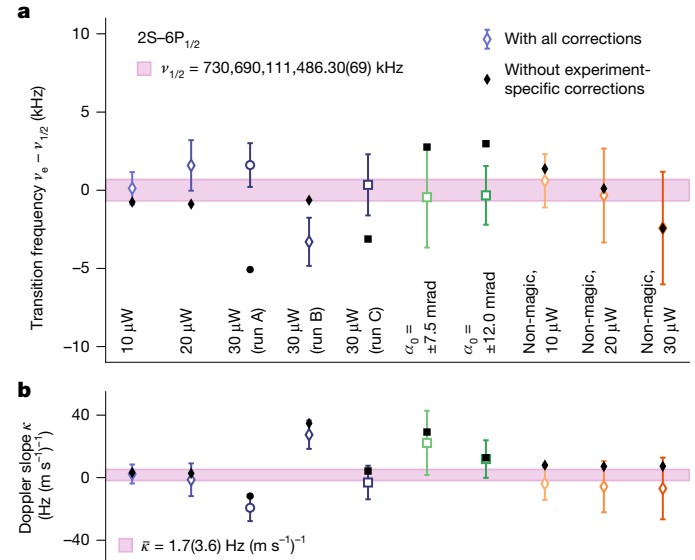

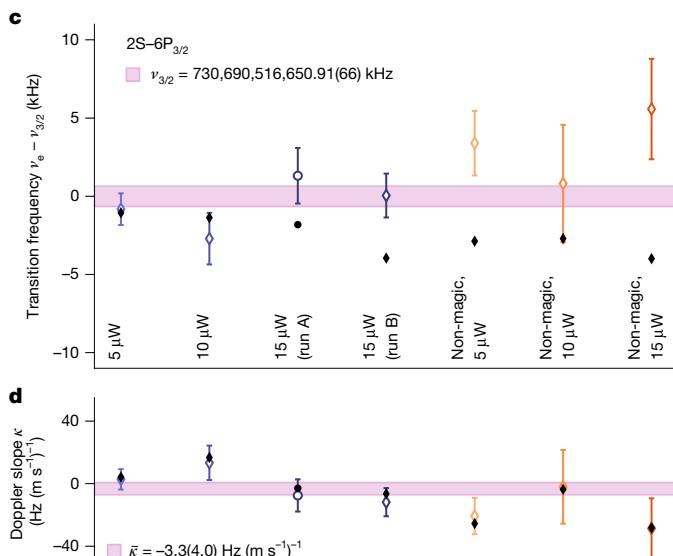

**Fig. 4 | Transition frequencies and Doppler slopes for the two examined 2S−6P transitions. a**, Measured transition frequencies $\nu_e$ of the 2S−6P$_{1/2}$ transition for different data groups (that is, combinations of experimental parameters and measurement runs), as labelled below each data point and listed in Extended Data Table 3. Data were taken for various combinations of spectroscopy laser power (10, 20 and 30 μW), atomic beam offset angle $\alpha_0$ ($\alpha_0 = 0$ mrad unless stated otherwise) and laser polarization angle $\theta_L$ (magic angle $\theta_L = 56.5°$ by default; non-magic angle $\theta_L = 146.5°$ marked accordingly). The data were collected over several months in 3-week-long runs, run A (circles), run B (diamonds) and run C (squares). Error bars show combined statistical and systematic one-standard-deviation ($\sigma$) uncertainty and the purple-shaded area shows the weighted mean of all data and its $1\sigma$ uncertainty. Black markers show the results without experiment-specific corrections, including for the light force, QI and first-order and second-order Doppler shifts (Extended Data Table 4). **b**, Doppler slopes $\kappa$ for the data groups of panel **a**, determined in situ using velocity-resolved detection (Fig. 2). The mean speed of the atomic beam is $\langle \overline{v} \rangle \approx 195$ m s$^{-1}$. $\overline{\kappa}$ is the weighted mean Doppler slope. **c,d**, Same as panels **a** and **b** but for the 2S−6P$_{3/2}$ transition. Spectroscopy laser powers have a 2:1 ratio for the 2S−6P$_{1/2}$ and 2S−6P$_{3/2}$ transitions to keep their Rabi frequencies identical. The Pearson correlation coefficient between $\nu_e$ and $\kappa$ is $r = -0.78$ over all data groups.

delocalized matter wave, with transverse velocity $\nu_x \approx \nu(\delta\alpha + \alpha_0)$. Figure 3a shows the LFS $\nu_{LFS}$, found with a Voigt line shape fit to the simulated line shape, for an atom with $\nu = 200$ m s$^{-1}$ (note that $\nu_{LFS}$ is always symmetric in $\nu_x$). The hypothetical situation without back decay to the 2S level (dotted black line) describes pure diffraction of the matter wave on the standing wave. At $\nu_x = \nu_{rec}$, the Bragg condition is met and the atoms coherently scatter photons between the two counterpropagating beams, resulting in a narrow resonance with shifts exceeding 200 kHz (Fig. 3b). Below this resonance, the shift is negative and approximately constant around zero $\nu_x$, and above the resonance, the shift is positive and tends to zero as $\nu_x$ increases. Allowing back decay, on the other hand, allows scattering of the atoms on the standing wave. This primarily leads to a positive shift above the resonance that approximately scales with $\gamma_{ei}/\Gamma$, as shown for the 2S−6P$_{1/2}$ ($\gamma_{ei}/\Gamma = 3.9\%$; blue solid line) and 2S−6P$_{3/2}$ ($\gamma_{ei}/\Gamma = 7.9\%$; dashed red line) transitions.

The partially coherent atomic beam corresponds to an incoherent sum over atoms in various momentum eigenstates (Supplementary Methods). Because the transverse velocity spread of the beam is on the order of $\nu_{rec}$ (see shaded area at the top of Fig. 3a showing the velocity distribution of selected velocity groups and Extended Data Table 2), this prevents us from experimentally resolving the Bragg resonance and partly averages out the shift (to at most −0.96 kHz and −1.28 kHz for the 2S−6P$_{1/2}$ and 2S−6P$_{3/2}$ transitions, respectively, both occurring for velocity group $\tau_{13}$).

To test our model and simulation of the LFS, we measured the transition frequency of the 2S−6P$_{1/2}$ transition at atomic beam offset angles of $\alpha_0 = \pm7.5$ mrad and $\alpha_0 = \pm12$ mrad, as well as the dataset with $\alpha_0 = 0$ mrad (Fig. 3c,d). In particular, we find the difference in Doppler-free transition frequency between the data taken at $\alpha_0 = \pm12$ mrad and $\alpha_0 = 0$ mrad (both for $P_{2S−6P} = 30$ μW), having applied a correction of 0.17(28) kHz for all systematic effects except the LFS, to be

$$\nu_e^{(\text{no LFS corr.})}(\pm12 \text{ mrad}) - \nu_e^{(\text{no LFS corr.})}(0 \text{ mrad}) = 4.32(1.83) \text{ kHz.} \quad (3)$$

The uncertainty is dominated by the statistical uncertainty of 1.63 kHz of the ±12 mrad data. This difference is in excellent agreement with the LFS simulation, which predicts $\nu_{e,LFS}(\pm12 \text{ mrad}) - \nu_{e,LFS}(0 \text{ mrad}) = 4.21(61)$ kHz. The accompanying distortion of the experimental line shape, clearly observable in the asymmetry of fit residuals (Extended Data Fig. 1), is distinctly different for the two values of $\alpha_0$ and in excellent agreement with the simulations. The measured LFS at $\alpha_0 = \pm7.5$ mrad is also in agreement with the simulation but has a comparatively large statistical uncertainty of 3.04 kHz (Fig. 3d). This comparison is also a powerful test of the Doppler shift suppression scheme (equation (3) includes a Doppler shift correction of $\Delta\nu_e = -1.92(1.81)$ kHz) because of the large difference in $\alpha_0$ and of the data analysis because of the qualitatively different line shapes involved.

## Quantum interference

We use the QI simulations developed and extensively tested in ref. 2 to estimate any remaining QI shift, which we find to be −0.25(36) kHz and 0.05(15) kHz for the 2S−6P$_{1/2}$ and 2S−6P$_{3/2}$ transitions, respectively (Methods). These values include a small subset of data taken orthogonal to the magic angle, at which QI distortions are larger, to test our simulations. Also, we make use of the fact that the distortions are of opposite sign (and different magnitude) for the two transitions[22] and combine the 2S−6P$_{1/2}$ and 2S−6P$_{3/2}$ transition frequencies with a 1:2 ratio into the 2S−6P fine-structure centroid (Methods). This reduces the shift to only −0.05(2) kHz.

## 2S−6P transition frequencies

By averaging all available data (Fig. 4), we find the transition frequencies of the 2S−6P$_{1/2}$ transition ($\nu_{1/2}$) and the 2S−6P$_{3/2}$ transition ($\nu_{3/2}$) to be

$$v_{1/2} = 730{,}690{,}111{,}486.30(49)_{\text{stat}}(49)_{\text{sys}} \text{ kHz}$$
$$= 730{,}690{,}111{,}486.30(69) \text{ kHz } [0.94 \text{ ppt}], \tag{4}$$

$$v_{3/2} = 730{,}690{,}516{,}650.91(60)_{\text{stat}}(28)_{\text{sys}} \text{ kHz}$$
$$= 730{,}690{,}516{,}650.91(66) \text{ kHz } [0.90 \text{ ppt}]. \tag{5}$$

The final one-standard-deviation uncertainties ($\sigma$) are the combined statistical ('stat') and systematic ('sys') uncertainties, with the former corresponding to the Doppler shift extrapolation uncertainty. All corrections and uncertainties are listed in Extended Data Table 4, with contributions not discussed in the main text detailed in Methods and Supplementary Methods. The results from the two detectors, averaged here, agree within their uncertainties. The average Doppler slopes $\bar{\kappa}$ of both transitions are compatible with zero.

By taking the difference of the two measured transition frequencies, we find the 6P fine-structure splitting between the $6P_{1/2}^{F=1}$ and $6P_{3/2}^{F=1}$ levels,

$$\Delta v_{\text{FS}}(6P) = v_{3/2} - v_{1/2} = 405{,}164.62(97) \text{ kHz}. \tag{6}$$

Extended Data Table 4 lists the corrections and uncertainties. The QED prediction $\Delta v_{\text{FS,QED}}(6P) = 405{,}164.51(1)$ kHz (Methods), which at our level of accuracy does not depend on $r_{\text{p}}$, is in excellent agreement ($\Delta v_{\text{FS}}(6P) - \Delta v_{\text{FS,QED}}(6P) = 0.11(97)$ kHz). This comparison tests the (uncorrelated) Doppler shift extrapolation and corrections for the light force, QI and dc-Stark shifts, with the last two of opposite sign for the two transitions.

Finally, we combine the two transition frequencies into the 2S–6P fine-structure centroid,

$$v_{2S-6P} = \frac{1}{3}v_{1/2} + \frac{2}{3}v_{3/2} + \Delta v_{\text{HFS}}(v_{2S-6P}) \tag{7}$$

$$v_{2S-6P} = 730{,}690{,}248{,}610.79(48) \text{ kHz } [0.66 \text{ ppt}], \tag{8}$$

with the hyperfine-structure (HFS) correction $\Delta v_{\text{HFS}}(v_{2S-6P}) = -132{,}985.25(1)$ kHz (Methods). The corrections and uncertainties for $v_{2S-6P}$ are listed in Table 1. The total applied correction, excluding the precisely known recoil shift and HFS corrections, corresponds to 3.6$\sigma$, with the largest individual correction (for the LFS) corresponding to 2.4$\sigma$. The relative uncertainty corresponds to a sixfold improvement over our previous 2S–4P measurement[2].

## $R_\infty$ and $r_{\text{p}}$ from atomic hydrogen

Combining the 2S–6P fine-structure centroid $v_{2S-6P}$ of equation (8) with a measurement of the 1S–2S transition frequency[18], and using equation (1), we determine the proton rms charge radius as

$$r_{\text{p}} = 0.8406(5)_{\text{QED}}(14)_{\text{exp}} \text{ fm} = 0.8406(15) \text{ fm}. \tag{9}$$

The total uncertainty arises mostly from the uncertainty of $v_{2S-6P}$ ('exp'), with the QED uncertainty of equation (1) ('QED') approximately threefold lower (contributions from the 1S–2S transition frequency and other physical constants are negligible). Figure 1 shows equation (9) with other relevant determinations of $r_{\text{p}}$. Equation (9) is the most precise value for $r_{\text{p}}$ from any measurement other than the muonic measurement[9,15], being 2.5-fold and sixfold more precise than the next best determination from atomic hydrogen[5] and our 2S–4P measurement[2], respectively. Equation (9) is in excellent agreement with the muonic measurement[15] ($r_{\text{p}} = 0.84060(39)$ fm) but disagrees with the CODATA 2014 proton radius[16] by 5.5$\sigma$. Instead of $r_{\text{p}}$, we may also determine the 1S Lamb shift as $\mathcal{L}_{\text{exp}}(1S) = 8{,}172{,}744.13(14)_{\text{QED}}(3.56)_{\text{exp}}$ kHz = 8,172,744.1(3.6) kHz (Methods).

**Table 1 | Corrections $\Delta v$ and uncertainties $\sigma$ for the determination of the 2S–6P fine-structure centroid $v_{2S-6P}$**

| Contribution | $\Delta v$ (kHz) | $\sigma$ (kHz) |
|---|---|---|
| First-order Doppler shift | 0.34 | 0.43 |
| – Extrapolation (statistical) | 0.34 | 0.43 |
| – Simulation of atom speeds | – | 0.01 |
| Simulation corrections | 1.05 | 0.17 |
| – LFS | 1.15 | 0.17 |
| – QI shift | 0.05 | 0.02 |
| – Second-order Doppler shift | −0.14 | 0.01 |
| dc-Stark shift | 0.05 | 0.07 |
| BBR-induced shift | 0.28 | 0.01 |
| Zeeman shift | 0.00 | 0.08 |
| Pressure shift | 0.00 | 0.02 |
| Sampling bias | 0.00 | 0.06 |
| Signal background | 0.00 | 0.03 |
| Laser spectrum | 0.00 | 0.07 |
| Frequency standard | 0.02 | 0.01 |
| Subtotal (experiment-specific contributions) | 1.74 | 0.48 |
| Recoil shift | −1,176.03 | 0.00 |
| HFS correction $\Delta v_{\text{HFS}}(v_{2S-6P})$ | −132,985.25 | 0.01 |
| Total (all contributions) | −134,159.54 | 0.48 |

All uncertainties correspond to one standard deviation. Indented entries detail subcontributions to the first-order Doppler shift and simulation corrections. The sum of the subcontributions may differ from the given total owing to rounding. BBR, blackbody radiation.

The Rydberg constant can be similarly extracted from the combination of the 1S–2S and 2S–6P transition frequencies, giving $R_\infty = 10{,}973{,}731.568152(14)$ m$^{-1}$ (1.3 ppt; Pearson correlation coefficient $r = 0.94$ with equation (9)). However, because the theory predictions of both frequencies depend on $R_\infty$ and $r_{\text{p}}$, we cannot make full use of the relative precision of 0.66 ppt of $v_{2S-6P}$.

## Test of the SM prediction

Having confirmed the muonic value of $r_{\text{p}}$, we now explicitly compare the SM prediction $v_{2S-6P,\text{SM}}$ for the 2S–6P fine-structure centroid with the experimentally determined value $v_{2S-6P}$. Combining the 1S–2S transition frequency[18] and the muonic value of $r_{\text{p}}$ (ref. 15) with equation (1) (Methods), we find

$$v_{2S-6P,\text{SM}} = 730{,}690{,}248{,}610.79(18)_{\text{QED}}(14)_{r_{\text{p}}} \text{ kHz}$$
$$= 730{,}690{,}248{,}610.79(23) \text{ kHz } [0.31 \text{ ppt}]. \tag{10}$$

The prediction is in excellent agreement with our experimental result with a difference of $v_{2S-6P} - v_{2S-6P,\text{SM}} = 0.00(53)$ kHz, corresponding to a test of the SM at a relative precision of 0.7 ppt. This precision is comparable with the test of the SM prediction of the electron magnetic moment, which is at present limited to 0.7 ppt by discrepant measurements of $\alpha$ (ref. 12). Likewise, we compare the 1S Lamb shift prediction $\mathcal{L}_{\text{QED}}(1S) = 8{,}172{,}744.1(1.3)_{\text{QED}}(1.0)_{r_{\text{p}}}$ kHz = 8,172,744.1(1.7) kHz with $\mathcal{L}_{\text{exp}}(1S)$, which, to our knowledge, is the most precise test (0.5 ppm) of bound-state QED corrections so far.

Extended Data Table 1 lists the QED corrections and their uncertainties. Our experimental uncertainty is comparable with the muonic and hadronic vacuum polarization corrections and reaches the level of three-photon corrections ($\propto \alpha^5$), matching the highest-order terms in the prediction of the electron magnetic moment[12]. Although the experimental uncertainty is 2.7-fold larger than the QED uncertainty (dominated by two-photon and radiative-recoil corrections), the latter is partly estimated by the expected size of uncalculated terms[1,38] as

opposed to experimental tests. In fact, a recent recalculation of the two-photon self-energy[39] would shift $\mathcal{L}_{QED}(1S)$ on the order of the experimental $\sigma$ of $\mathcal{L}_{exp}(1S)$.

Equivalently, we can compare the values of $R_\infty$ determined from $\nu_{2S-6P}$ or the 1S–2S transition frequency, each in combination with the muonic value of $r_p$ and equation (1). For $\nu_{2S-6P}$, this gives

$$R_\infty = 10{,}973{,}731.5681524(25)_{QED}(72)_{exp}(19)_{r_p} \text{ m}^{-1} \tag{11}$$
$$= 10{,}973{,}731.5681524(79) \text{ m}^{-1} [0.75 \text{ ppt}],$$

with the uncertainty limited by the precision of $\nu_{2S-6P}$. Using the 1S–2S transition frequency yields a compatible $R_\infty = 10{,}973{,}731.5681523(65)$ m$^{-1}$ ($r_{1S-2S} = 0.40$ correlation with equation (11)), limited by the uncertainty of $\mathcal{L}_{QED}(1S)$. Equation (11) is compatible with and 50% more precise than the $R_\infty$ world average[1], which has an expanded uncertainty accounting for scatter in the input data. Adding the 2S–6P measurement will probably reduce scatter in future adjustments as less precise measurements are removed. This situation is illustrated in Extended Data Fig. 2. Our measurement can also be used to improve new physics constraints on weakly interacting bosons with masses in the keV range[40,41].

The techniques demonstrated here may be used with any 2S–$n$P transitions in atomic hydrogen and deuterium[42], with a precision matching or exceeding that of SM predictions feasible. Together with complementary approaches[43–47], we expect this to substantially advance bound-state QED tests.

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

## Methods

### Data acquisition

Here we give further relevant details on the experimental scheme and apparatus (both described in detail in ref. 23). The run time of the cryogenic atomic beam is limited to freezing cycles of approximately 2 h by the accumulation of frozen (molecular) hydrogen inside the nozzle, which is removed by heating the nozzle to room temperature. The vacuum ($2 \times 10^{-7}$ mbar, dominated by molecular hydrogen) inside the 2S–6P spectroscopy region is maintained by differential pumping with a cryopump to minimize pressure shifts[50–52], with the temperature of the apparatus allowed to equilibrate on each measurement day before collecting data.

The power of the linearly polarized, 243-nm 1S–2S preparation laser[53,54] is resonantly enhanced in an in-vacuum, standing-wave cavity to $P_{\text{1S–2S}} \approx 1$ W per direction (297 μm $1/e^2$ intensity waist radius). The observed 1S–2S transition linewidth is approximately 3 kHz (FWHM; atomic detuning, as opposed to laser detuning), limited by single-photon ionization of the 2S level[55]. Therefore, the $2S_{1/2}^{F=1}$ levels are only populated by Doppler-sensitive two-photon excitation, leading to a population of approximately $7 \times 10^{-7}$ in each sublevel relative to the population in the $2S_{1/2}^{F=0}$ level. The detuning of the preparation laser is set several times per freezing cycle by observing the 1S–2S transition. An equal-slit-width optical chopper running at 160 Hz periodically blocks the preparation laser (which sets delay time $\tau = 0$ μs) to enable the velocity-resolved detection. The channel electron multipliers are switched off with a fast high-voltage switch while the preparation laser is unblocked to prevent saturation from scattered 243-nm light.

By using a linearly polarized 2S–6P spectroscopy laser[53,54] and the $2S_{1/2}^{F=0}$ level as the initial level, only transitions to the $6P_{1/2}^{F=1}$ hyperfine level (2S–6P$_{1/2}$ transition) and the $6P_{3/2}^{F=1}$ hyperfine level (2S–6P$_{3/2}$ transition) are dipole-allowed, whereas the excitation of the $6P_{1/2}^{F=0}$ and $6P_{3/2}^{F=2}$ levels is prevented by angular momentum conservation (Fig. 2a). The line strengths ($\propto \mu^2$, in which $\mu$ is the dipole moment) of the 2S–6P$_{1/2}$ and 2S–6P$_{3/2}$ transitions have a 1:2 ratio. We use spectroscopy laser powers $P_{\text{2S–6P}}$ with a ratio of 2:1 for the two transitions to keep their Rabi frequencies $\Omega_0 \propto \mu \sqrt{P_{\text{2S–6P}}}$ identical (the peak Rabi frequency is $(2\pi \times 126)$ krad s$^{-1}$ at our highest spectroscopy laser powers of 30 and 15 μW, respectively). Most 6P decays (branching ratio $\Gamma_{\text{e–1S}}/\Gamma = 88.2\%$) are Lyman decays to the 1S manifold, with the most energetic, direct Lyman-ε decay ($\Gamma_{\text{det}}/\Gamma = 80.5\%$) dominating, whereas the remaining $\gamma_{\text{e–2S}}/\Gamma = 11.8\%$ are Balmer decays to the 2S manifold, of which in turn a fraction $\gamma_{\text{ei}}/\Gamma$ leads back to the initial $2S_{1/2}^{F=0}$ level (see Section 1.2 in the Supplementary Methods). The metastable 2S levels are treated as stable here, as their natural lifetime (122 ms) is much longer than the time the atoms spend in the atomic beam (see Section 2.6 in the Supplementary Methods for the 2S decay contribution to the signal background).

The channel electron multipliers count the fluorescence photons from the 6P decays, either by detecting the photoelectrons emitted by the photons from the detector cylinder walls or, to a lesser extent, directly detecting the photons. Because the photoelectron yield strongly increases with photon energy (for both colloidal graphite and oxidized aluminium, the surface materials of the Faraday cage and the detector cylinder, respectively), fluorescence from Lyman-ε decay (13.2 eV photon energy) constitutes approximately 97% of the signal detected by the channel electron multipliers. The counts are binned into the 16 velocity groups by their delay time $\tau$, with the bins chosen to cover a wide range of mean speeds $\bar{v}$ while exhibiting a sufficient signal-to-noise ratio (bin width 50–550 μs; Extended Data Table 2). For each line scan, we accumulate counts over 160 chopper cycles at each spectroscopy laser detuning. Intermittently, excess scatter and spiking was observed for the bottom detector and its signal was subsequently discarded (≈11% of line scans; Extended Data Table 3). We attribute this to the bottom detector being cooled down to close to, and possibly

below, its lower operating temperature limit because of its vicinity to the cryopump.

At least once per measurement day, the nozzle and the collimating aperture are centred on the preparation laser. At the start of each freezing cycle, the atomic beam offset angle $\alpha_0$, which is controlled by a linear motor equipped with a position sensor, is aligned to zero with a 1 mrad alignment uncertainty. This is achieved by blocking the returning beam of the spectroscopy laser (using an in-vacuum shutter in front of the high-reflectivity mirror of the AFR[25,26]), determining the (now unsuppressed) Doppler slope $\kappa$ for several angles and moving to the angle at which $\kappa$ is zero, which is set as $\alpha_0 = 0$ mrad. To record line scans at a non-zero angle $\pm\alpha_0$, we first move to either $+\alpha_0$ or $-\alpha_0$ (chosen randomly) and then to the opposite sign, recording typically 5–10 scans at each angle, and repeating this procedure several times per freezing cycle. The fibre–collimator distance of the AFR is optimized[25,26] at least once per freezing cycle.

We use fixed sets of 30 symmetric (15 unique) detunings $\Delta$ of the spectroscopy laser frequency to sample the 2S–6P fluorescence line shape, with different sets used for $\alpha_0 = 0$, $\pm 7.5$ and $\pm 12$ mrad to account for the different line shapes (see Section 5.1.3 and Table 5.2 of ref. 23), all of which have $\pm 50$ MHz as the largest detuning. The detunings were chosen to minimize the statistical uncertainty of the Doppler-free transition frequency $\nu_e$, whereas the number of detunings and the acquisition time (1 s) at each detuning were chosen to balance between sufficient line sampling, signal-to-noise ratio and number of line scans per freezing cycle. For each line scan, the order of the detunings is randomized to minimize the influence of drifts in the signal. At the beginning of each freezing cycle, the centre laser frequency (to which the detunings are added) was chosen randomly from a normal distribution. This distribution was centred on the 2S–6P transition frequency expected from the muonic measurement of the proton radius[9], with a standard deviation of 12 kHz to cover the transition frequency expected from the CODATA 2014 value of the proton radius[16]. The laser frequencies are referenced to the caesium frequency standard using an optical frequency comb[56,57] and a global navigation satellite system (GNSS)-referenced hydrogen maser (see Section 2.7 in the Supplementary Methods).

We switched between examining the 2S–6P$_{1/2}$ and 2S–6P$_{3/2}$ transitions several times during the measurement, except during measurement run C, for which only the 2S–6P$_{1/2}$ transition was examined for several values of $\alpha_0$ (Extended Data Table 3).

### Voigt and Voigt doublet line shapes

A Voigt line shape[23,58] is the convolution of a Lorentzian line shape (with FWHM linewidth $\Gamma_{\text{L}}$) and a Gaussian line shape (with FWHM linewidth $\Gamma_{\text{G}}$). It has a combined FWHM linewidth $\Gamma_{\text{F}} \approx 0.5346\Gamma_{\text{L}} + \sqrt{0.2166\Gamma_{\text{L}}^2 + \Gamma_{\text{G}}^2}$ (ref. 59) and amplitude $A$ and is centred on the resonance frequency $\nu_0$. A constant offset $y_0$ is added to account for the experimentally observed signal background, resulting in five free parameters.

The Voigt doublet line shape is here defined as the sum of two Voigt line shapes with generally different resonance frequencies $\nu_1$ and $\nu_2$ and amplitudes $A_1$ and $A_2$ but equal Lorentzian and Gaussian linewidths[23] (and a constant offset $y_0$), resulting in seven free parameters. Its resonance frequency is the centre of mass of the constituent line shapes, that is, $\nu_0 = (A_1\nu_1 + A_2\nu_2)/(A_1 + A_2)$.

### Data analysis

The resonance frequency $\nu_0$ of each velocity group and detector of each line scan is determined by least-square fitting Voigt line shapes to the signal (for data with $\alpha_0 = 0$ mrad, that is, data groups G1A–G12). For data with $\alpha_0 \neq 0$ mrad (data groups G13 and G14), in which the line can split into two Doppler components, the Voigt doublet line shape is used instead. For the fits, we assume the uncertainty on the signal $y_j$, that is, the number of fluorescence photons detected at detuning $j$, to be the photon-number shot (Poissonian) noise, given by $\sqrt{y_j}$

(because $y_j \gg 1$, the Poisson distribution can be approximated with a normal distribution).

We use reduced chi-squared $\chi^2_{\text{red}} = \chi^2/k$ (with $k$ degrees of freedom) as a measure of the goodness of fit. The average $\chi^2_{\text{red}}$ is expected to be 1 if the fitted line shapes exactly match the experimental line shapes and if the signal fluctuations are fully described by shot noise. We first discuss the Voigt fits ($k = 25$), for which we find $\chi^2_{\text{red}}$ to be substantially larger than 1 (up to $\chi^2_{\text{red}} \approx 4.6$ at our highest spectroscopy laser power) for all but the slowest velocity groups. Only for those velocity groups, which have a comparatively lower signal and Doppler broadening, does $\chi^2_{\text{red}}$ approach 1 ($\chi^2_{\text{red}} \approx 1.1$ for $\tau_{16}$ for all data groups).

We identify two distinct contributions to the increased $\chi^2_{\text{red}}$. First, there are small model deviations between the fitted and experimental line shapes, caused by non-Gaussian (and non-Lorentzian) broadening and saturation effects not included in the former and clearly observable in the fit residuals (up to approximately 2% deviation; see black circles at the top of Extended Data Fig. 1b). Our simulated line shapes (both LFS and QI simulations) include these effects and consequently show much better agreement with the experimental line shapes (see dashed and solid lines in Extended Data Fig. 1b). We use this to determine the effect of the deviations on $\chi^2_{\text{red}}$ by repeating the line shape fits using simulated data instead of experimental data. In this Monte Carlo simulation, the appropriate line shape simulation is scaled and offset to match the experimental data of each line scan and then shot noise is added to the simulated signal. The simulated $\chi^2_{\text{red}}$ values reproduce, and therefore the deviations explain, approximately 70% of the excess (that is, $\chi^2_{\text{red}} - 1$) in the experimental $\chi^2_{\text{red}}$ values.

Second, there are fluctuations in the signal that lead to excess (technical) noise above shot noise, which we attribute to fluctuations in the atomic flux (of metastable 2S atoms). In particular, we identify correlations between $\chi^2_{\text{red}}$ of a line scan and fluctuations of the nozzle temperature (Pearson correlation coefficient $r = 0.45$) and the preparation laser power ($r = 0.26$) during the line scan, both of which directly affect the number of 2S atoms reaching the spectroscopy region. The correlation with the spectroscopy laser power is much smaller ($r = 0.03$). We find that the simple assumption of a velocity-group-independent, 1% rms fluctuation of the signal from detuning to detuning explains, on average, the remaining excess of the experimental $\chi^2_{\text{red}}$ values.

The $\chi^2_{\text{red}}$ behaviour is very similar for the Voigt doublet fits ($k = 23$) to the $\alpha_0 = \pm 7.5$ mrad data (data group G13). For the $\alpha_0 = \pm 12$ mrad data (data group G14), the two Doppler components start to separate, particularly for fast velocity groups (Fig. 3c). This leads to large model deviations (up to 14%; see bottom of Extended Data Fig. 1b) from saturation effects, as some atoms interact with both spectroscopy laser beams ($|\Delta| \lesssim \Gamma_{2P}$), whereas others only interact with one beam ($|\Delta| \gtrsim \Gamma_{2P}$). Consequently, $\chi^2_{\text{red}}$ can exceed 20, which is largely explained (90% of excess) by the deviations, as again determined from the Monte Carlo simulation.

Asymmetric (about the line centre) deviations between the fitted and experimental line shapes, arising as a result of LFS and QI, do not substantially influence $\chi^2_{\text{red}}$ because of their small size. However, both symmetric and asymmetric deviations lead to a sampling bias, as discussed in Section 2.5 in the Supplementary Methods.

The uncertainty of the experimental resonance frequency $\nu_0$ is estimated from the line shape fits assuming only shot noise, that is, the technical noise is not taken into account at this point. The value of $\nu_0$ for each velocity group and detector of each line scan is corrected for the LFS and QI shift by subtracting, respectively, $\nu_{\text{LFS}}$ and $\nu_{\text{QI}}$, which are determined from the corresponding simulated line shapes (see Section 1.1 in the Supplementary Methods). The mean speed $\bar{\nu}$ and rms speed $\bar{\nu}_{\text{rms}}$ of each velocity group are also determined from these simulated line shapes (see Section 1.1 in the Supplementary Methods; Extended Data Table 2 gives the average value of $\bar{\nu}$ for each velocity group). The second-order Doppler shift $\Delta\nu_{\text{SOD}}$ is calculated using $\bar{\nu}_{\text{rms}}$ (see Section 2.1 in the Supplementary Methods) and likewise subtracted from $\nu_0$.

Next, using the simulated values of the speed $\bar{\nu}$, the Doppler shift extrapolation is performed to find the Doppler-free transition frequency $\nu_e$ and the Doppler slope $\kappa$. The values $\nu_e$ and $\kappa$ found for each line scan are inherently strongly correlated (Pearson correlation coefficient $r_e \approx -0.97$) and their uncertainties are found by propagating the uncertainties of $\nu_0$. Although the averaging process outlined below reduces this correlation, the correlation remains substantial ($r = -0.78$) between the averaged values of $\nu_e$ and $\kappa$ of different data groups. The LFS, QI shift and second-order Doppler shift all depend on the speed of the 2S atoms, either indirectly through the interaction time or directly. This results in non-zero Doppler slopes if not corrected for. In particular, the LFS, on average, would lead to $\kappa \approx -5$ Hz (m s$^{-1}$)$^{-1}$ and it is only by accounting for it that the experimentally determined values of $\kappa$ are, on average, compatible with zero.

The $\chi^2_{\text{red}}$ of the Doppler shift extrapolation ($k = 14$) is close to 1 ($\chi^2_{\text{red}} = 1.04(2)$ on average; standard deviation over data groups in parentheses), showing that the data are well described by a linear model and technical noise is small compared with shot noise. This contrasts with the excess noise observed in the line shape fits, as discussed above. We attribute this to the different timescales involved: the different velocity groups are recorded for about 100 μs and within 2,560 μs of each other, whereas the signal at each detuning is accumulated for 1 s before moving to the next detuning. That is, the Doppler shift extrapolation is mainly susceptible to technical noise on timescales of 100 μs, whereas the signal at different detunings is susceptible to technical noise on timescales of about 1 s, which is, for example, the timescale expected for nozzle temperature fluctuations.

Next we find the weighted mean of $\nu_e$ from the two detectors for each line scan (except when no data from the bottom detector are available, in which case data from the top detector are used), taking into account experimentally determined correlations ($r = 0.36(15)$ on average). We attribute these correlations, which tend to increase with spectroscopy laser power and therefore the signal, again to the detuning-to-detuning fluctuations in the atomic flux that are common mode to both detectors (and the velocity groups). We then form the weighted mean of the detector-averaged $\nu_e$ for each freezing cycle in a given data group. The reduced chi-squared $\chi^2_{\text{red,FC}}$ of this average is typically greater than 1 ($\bar{\chi}^2_{\text{red,FC}} = 1.44(23)$ on average), which we attribute again to fluctuations in the atomic flux. We account for this excess scatter by scaling the uncertainty of $\nu_e$ by the corresponding $\sqrt{\chi^2_{\text{red,FC}}}$ if $\chi^2_{\text{red,FC}} > 1$. This procedure shifts the determined transition frequencies (by changing the weighting of the data) by less than 20 Hz, much smaller than the associated uncertainties. A weighted mean of the detector-averaged, uncertainty-scaled $\nu_e$ is formed for each data group and the remaining corrections are applied (Table 1 and Extended Data Table 4). Finally, the relevant data groups are (weighted) averaged to find the transition frequencies of the 2S–6P$_{1/2}$ and 2S–6P$_{3/2}$ transitions. Throughout the analysis, the weights of the averages are based only on the (scaled) statistical frequency uncertainty (including for the Doppler slope $\kappa$), with the (correlated) uncertainty of the corrections not included in the weights. The statistical weights $w_{2S-6P}$ of the data groups for the determination of the 2S–6P fine-structure centroid $\nu_{2S-6P}$ are given in Extended Data Table 3.

The data analysis was blinded by adding a randomly chosen offset frequency to the transition frequencies. The offset frequency was only removed after all of the main systematic effects had been studied. Further small corrections, identified after the offset frequency was removed, resulted in a negligible shift of the determined transition frequencies of at most 10 Hz. The results of the data analysis (performed by L.M.) were confirmed by a second, independently implemented analysis (performed by V.W.).

## Modelling of atomic beam and fluorescence line shape

The fluorescence line shape of the 2S–6P transition is modelled by a Monte Carlo simulation of the atomic beam as a set of atomic

trajectories, the trajectories' interaction with the 1S–2S preparation and 2S–6P spectroscopy lasers and their contribution to the fluorescence signal. The procedure is described in detail in Section 1.1 in the Supplementary Methods. Two complementary models describe the interaction with the spectroscopy laser, the QI model (see below and Section 1.2 in the Supplementary Methods) and the LFS model (see main text and Section 1.3 in the Supplementary Methods).

## Speed distribution of atomic beam

We use the signal of the velocity groups as a time-of-flight measurement of the speed distribution of the atomic beam. To this end, we compare the experimental values of the line amplitudes $A$ of the velocity groups to the values of $A$ of line shapes simulated using a given speed distribution (see Section 1.1 in the Supplementary Methods). We find that the probability distribution of the speed $v$ of atoms leaving the nozzle towards the 2S–6P spectroscopy region is well described by a modified Maxwell–Boltzmann flux distribution for a wide range of experimental parameters (see ref. 23 for details). The flux distribution is given by

$$p_{\text{eff}}(v) = \mathcal{N} v^3 e^{-\frac{m_H v^2}{2 k_B T_N}} e^{-\frac{v_{\text{cut-off}}}{v}}, \tag{12}$$

in which $\mathcal{N}$ is a normalization constant. The extra factor $\exp(-v_{\text{cut-off}}/v)$ accounts for the depletion of slower atoms through collisions inside the nozzle, inside the beam and with the background gas[60–62]. A similar depletion has been observed in our 2S–4P measurement and other atomic hydrogen beams[3,63].

Using the above comparison of experimental data with simulations, $v_{\text{cut-off}}$ is found to be 50 m s$^{-1}$ on average. Extended Data Table 3 lists the average value and variation for each data group. A substantial part of the variation is because of the fact that $v_{\text{cut-off}}$ typically increases during a freezing cycle (see Fig. 6.1 of ref. 23). This is because the accumulation of frozen hydrogen decreases the diameter of the nozzle, causing an increase in the gas pressure and therefore collisions inside the nozzle. We take this variation into account when determining (the uncertainty of) the mean speeds of the velocity groups and the simulation corrections (see Section 1.1 in the Supplementary Methods and Extended Data Table 5).

## QI shift

We simulate QI-distorted line shapes with a model combining optical Bloch equations with simulations of the spatial detection efficiency, averaged over a set of trajectories representing the atomic beam (see Sections 1.1 and 1.2 in the Supplementary Methods). The QI shift is here defined as the centre frequency of a line shape fit to the simulated line shapes (see Section 1.1 in the Supplementary Methods). The validity of this approach was demonstrated by the excellent agreement between the observed and simulated QI shifts in our previous measurement of the 2S–4P transition[2], in which the shifts were more than sevenfold larger because of the smaller detection solid angle and larger linewidth[22].

The simulated QI shifts, as a function of polarization angle $\theta_L$, are found to be at most 7.1 kHz and −4.1 kHz for the 2S–6P$_{1/2}$ and 2S–6P$_{3/2}$ transitions at our highest spectroscopy laser power (used to give upper limits here and below), respectively. At the magic angle $\theta_L = 56.5°$, the shifts reduce to at most −0.87(54) kHz and 0.45(27) kHz (including ±3° alignment uncertainty; again at the highest spectroscopy laser power). When we account for data taken at $\theta_L = 146.5°$ (see below) and at lower laser powers (with statistical weights as given in Extended Data Table 3), we obtain the overall simulated shifts (−0.25(36) kHz and 0.05(15) kHz) given in the main text. The cancellation inherent in the 2S–6P fine-structure centroid reduces the shift to at most −0.37 kHz at any polarization angle and to below 0.01 kHz at around $\theta_L = 56.5°$ (below the otherwise negligible ac-Stark shift; see Section 1.2 in the Supplementary Methods). Including all data results in the shift of −0.05(2) kHz given in the main text.

The magic angle $\theta_L = 56.5°$ used in the measurement was determined with simulations before the measurement began, whereas more refined simulations of the spatial detection efficiency (completed after the measurement and used for all simulation results given here) result in a magic angle of approximately 52°. Moreover, the magic angle also slightly shifts with laser power (by up to 2° for the powers used here; see Section 1.2 in the Supplementary Methods). However, the QI shifts are still strongly suppressed at the magic angle used, despite it being slightly different from the optimal value.

To test our QI model and simulations, a limited amount of data were taken at $\theta_L = 146.5°$ (Fig. 4 and Extended Data Table 3), that is, orthogonal to the magic polarization angle, which we compare with the data taken at $\theta_L = 56.5°$. For the 2S–6P$_{1/2}$ transition, the difference in Doppler-free transition frequency of $v_e(\theta_L = 146.5°) - v_e(\theta_L = 56.5°) = -0.01(1.69)$ kHz is in excellent agreement with 0 after correcting for a differential QI shift of 3.44(92) kHz and a differential Doppler shift of −1.42(1.42) kHz. We may also compare the experimental and simulated line shape distortions at $\theta_L = 146.5°$, in which the QI shift dominates over the LFS, by comparing the asymmetry of the experimental and simulated fit residuals (Extended Data Fig. 1). We find excellent agreement, especially at detunings larger than the linewidth, for which the line shape distortions from QI are largest.

For the 2S–6P$_{3/2}$ transition, we find a moderate (2.3 standard deviations) tension with a difference of $v_e(\theta_L = 146.5°) - v_e(\theta_L = 56.5°) = 4.08(1.77)$ kHz in the transition frequency, having corrected for a differential QI shift of −1.78(47) kHz. The removed differential Doppler shift is likewise significantly non-zero (−3.66(1.68) kHz). This correlation is consistent with, but not conclusive evidence for, the non-zero difference being caused by random errors affecting the Doppler shift extrapolation. This conclusion is also supported by the fact that there is no tension in the velocity-group-averaged resonance frequency, that is, assuming zero Doppler shift (see Fig. 6.9 of ref. 23). Furthermore, we find the line shape distortions at $\theta_L = 146.5°$ to be compatible with our QI simulations but incompatible with a QI shift of opposite sign and twice the magnitude as implied by the measured difference (Extended Data Fig. 1). We therefore conclude that the tension probably results from random errors in the determination of the resonance frequencies.

## 2S–6P fine-structure centroid, 6P fine-structure splitting and HFS corrections

The 2S–6P fine-structure centroid $v_{2S–6P}$ is the transition frequency from the 2S HFS centroid to the 6P fine-structure centroid. It is determined from the two measured transition frequencies $v_{1/2}$ and $v_{3/2}$ for the transitions from the 2S$_{1/2}^{F=0}$ level to the 6P$_{1/2}^{F=1}$ level (2S–6P$_{1/2}$ transition) and the 6P$_{3/2}^{F=1}$ level (2S–6P$_{3/2}$ transition), respectively, by first correcting $v_{1/2}$ and $v_{3/2}$ for the 2S and 6P HFS and then averaging the corrected $v_{1/2}$ and $v_{3/2}$ weighted by their fine-structure multiplicity ratio of 1:2 (equivalent to the ratio of the line strengths $\propto \mu^2$ of the 2S–6P$_{1/2}$ and 2S–6P$_{3/2}$ transitions, in which $\mu$ is the dipole moment) to find the 6P fine-structure centroid. This results in equation (7), with the HFS corrections included in $\Delta v_{\text{HFS}}(v_{2S–6P})$, as detailed below.

The hyperfine interaction splits the fine-structure levels 2S$_{1/2}$, 6P$_{1/2}$ and 6P$_{3/2}$ into doublets[64,65]. The relevant HFS levels 2S$_{1/2}^{F=0}$, 6P$_{1/2}^{F=1}$ and 6P$_{3/2}^{F=1}$ are shifted from the fine-structure levels by the HFS energies $\Delta v_{\text{HFS}}(2S_{1/2}^{F=0})$, $\Delta v_{\text{HFS}}(6P_{1/2}^{F=1})$ and $\Delta v_{\text{HFS}}(6P_{3/2}^{F=1})$, respectively (see Fig. 6.11 of ref. 23 for the relevant level scheme). The value of $\Delta v_{\text{HFS}}(2S_{1/2}^{F=0})$ and its uncertainty are obtained from a measurement[66] of the 2S HFS splitting $\Delta v_{\text{HFS}}(2S_{1/2})$ as

$$\Delta v_{\text{HFS}}(2S_{1/2}^{F=0}) = -(3/4)\Delta v_{\text{HFS}}(2S_{1/2}) = -133,167,625.7(5.0) \text{ Hz.} \tag{13}$$

$\Delta v_{\text{HFS}}(6P_{1/2}^{F=1})$ and $\Delta v_{\text{HFS}}(6P_{3/2}^{F=1})$ can be calculated as detailed in refs. 64,65. They include small corrections from off-diagonal elements in the HFS Hamiltonian, which mix HFS levels with the same value of $F$ but different values of $J$. Because only the $F = 1$ level of each HFS doublet

is shifted by this effect, the centres of gravity of the HFS doublets are shifted by $\Delta\nu_{\mathrm{HFS}}^{\mathrm{o.d.}}(6P_{1/2})$ and $\Delta\nu_{\mathrm{HFS}}^{\mathrm{o.d.}}(6P_{3/2})$. Using the values for the 6P HFS splittings $\Delta\nu_{\mathrm{HFS}}(6P_{1/2})$, $\Delta\nu_{\mathrm{HFS}}(6P_{3/2})$ and the values for $\Delta\nu_{\mathrm{HFS}}^{\mathrm{o.d.}}(6P_{1/2})$, $\Delta\nu_{\mathrm{HFS}}^{\mathrm{o.d.}}(6P_{3/2})$ given in Table 1 of ref. 64 (see also equations (29) and (30) in ref. 65 and comments therein), we find

$$\Delta\nu_{\mathrm{HFS}}(6P_{1/2}^{F=1}) = (1/4)\Delta\nu_{\mathrm{HFS}}(6P_{1/2}) + \Delta\nu_{\mathrm{HFS}}^{\mathrm{o.d.}}(6P_{1/2}) = 547{,}798(6) \text{ Hz}, \quad (14)$$

$$\Delta\nu_{\mathrm{HFS}}(6P_{3/2}^{F=1}) = -(5/8)\Delta\nu_{\mathrm{HFS}}(6P_{3/2}) + \Delta\nu_{\mathrm{HFS}}^{\mathrm{o.d.}}(6P_{3/2})$$
$$= -547{,}460(6) \text{ Hz}. \quad (15)$$

$\Delta\nu_{\mathrm{HFS}}(6P_{1/2}^{F=1})$ and $\Delta\nu_{\mathrm{HFS}}(6P_{3/2}^{F=1})$ are assumed to be fully correlated.

The resulting HFS correction of the 2S–6P fine-structure centroid $\nu_{2S-6P}$ is

$$\Delta\nu_{\mathrm{HFS}}(\nu_{2S-6P}) = -(1/3)\Delta\nu_{\mathrm{HFS}}(6P_{1/2}^{F=1})$$
$$-(2/3)\Delta\nu_{\mathrm{HFS}}(6P_{3/2}^{F=1}) + \Delta\nu_{\mathrm{HFS}}(2S_{1/2}^{F=0}) \quad (16)$$
$$= -132{,}985{,}250(10) \text{ Hz},$$

in which we rounded to the nearest 10 Hz, as done for all corrections.

## SM predictions of transition frequencies and bound-state QED test

We find the SM prediction $\nu_{2S-6P,\mathrm{SM}}$ of the 2S–6P fine-structure centroid using

$$\nu_{2S-6P,\mathrm{SM}} = \frac{(1/3)E_{6,1,1/2} + (2/3)E_{6,1,3/2} - E_{2,0,1/2}}{E_{2,0,1/2} - E_{1,0,1/2}}\nu_{1S-2S}$$
$$= \frac{E(2S-6P)}{E(1S-2S)}\nu_{1S-2S}, \quad (17)$$

in which $E_{nlj}$ are the fine-structure level energies from equation (1) and $\nu_{1S-2S}$ is the measured frequency of the 1S–2S hyperfine centroid[18]. This parametrization removes the explicit dependence of $E_{nlj}$ on $R_\infty$. Using the muonic value of $r_\mathrm{p}$ (ref. 15), we find the value of $\nu_{2S-6P,\mathrm{SM}}$ given in equation (10).

Extended Data Table 1 lists the contributions to $\nu_{2S-6P,\mathrm{SM}}$. Along with the Dirac eigenvalue ($cR_\infty f_{nlj}^{\mathrm{Dirac}}$ of equation (1); equation (30) in ref. 1), we list the individual QED corrections ($cR_\infty f_{nlj}^{\mathrm{QED}}$ and $cR_\infty\delta_{l0}(C_{\mathrm{NS}}/n^3)\,r_\mathrm{p}^2$ of equation (1); equations (32)–(64) in ref. 1), with the sum of the corrections corresponding to the Lamb shift $\mathcal{L}(\mathcal{L}(nlj))$ for a single fine-structure level; $\mathcal{L}(2S-6P) = (1/3)\mathcal{L}(6,1,1/2) + (2/3)\mathcal{L}(6,1,3/2) - \mathcal{L}(2,0,1/2)$ for the 2S–6P fine-structure centroid). Extended Data Table 1 also gives the QED-only uncertainty, which we define as the uncertainty excluding contributions from $r_\mathrm{p}$, $\alpha$, $m_\mathrm{p}/m_\mathrm{e}$ and $\nu_{1S-2S}$. All listed QED corrections scale as $cR_\infty C/n^3$ in leading order for S-states, in which $cR_\infty C$ is the corresponding leading-order QED correction to the 1S level. This includes the (leading order, $\propto\alpha^2\times\alpha^2$, with the first $\alpha^2$ factor absorbed in $R_\infty = \alpha^2 m_\mathrm{e}c/2h$ in equation (1)) nuclear size correction highlighted in equation (1), for which $C = C_{\mathrm{NS}}r_\mathrm{p}^2$. Therefore, the corrections between different S-states are correlated, which is taken into account in the uncertainty of $\nu_{2S-6P,\mathrm{SM}}$. Non-S-states have generally much smaller corrections (for example, the fractional corrections are $1.3\times10^{-6}$ of the 2S binding energy but $4.4\times10^{-9}$ of the 6P binding energy) and, in particular, their nuclear size corrections are at present negligible (at most $7\times10^{-16}$ of the 6P binding energy). There is no correlation between the different QED corrections and we find the total QED-only uncertainty of 179 Hz by adding the uncertainties of the corrections in quadrature. Overall, $E(2S-6P)$ and $E(1S-2S)$ are highly correlated ($r = 0.995$ if only considering QED-only uncertainty).

The other dominant source of uncertainty for $\nu_{2S-6P,\mathrm{SM}}$ is the muonic value of $r_\mathrm{p}$, which contributes 138 Hz to the uncertainty of $\nu_{2S-6P,\mathrm{SM}}$. Although the uncertainty of $r_\mathrm{p}$ itself partly originates from QED corrections[15], the QED predictions for the energy levels of hydrogen and muonic hydrogen are uncorrelated at the present level of accuracy[1] and we therefore treat the QED-only uncertainty and the uncertainty from $r_\mathrm{p}$ as uncorrelated. The uncertainty contributions from $\alpha$ (1.7 Hz), $m_\mathrm{p}/m_\mathrm{e}$ (0.1 mHz) and $\nu_{1S-2S}$ (3 Hz) are negligible. In total, this results in an uncertainty of 226 Hz on $\nu_{2S-6P,\mathrm{SM}}$.

It is instructive to find the sensitivity of a prediction $\nu_{\mathrm{SM}}$, derived in the same way as $\nu_{2S-6P,\mathrm{SM}}$, to changes in $C$. Making use of the $1/n^3$ scaling of the QED corrections (and using the approximation $E_{nlj}\approx chR_\infty(-1/n^2 + \delta_{l0}C/n^3)$, that is, ignoring all non-leading-order corrections to $f_{nlj}^{\mathrm{Dirac}}$ and all QED corrections except $C$), we find

$$\frac{\partial}{\partial C}\left(\frac{\nu_{\mathrm{SM}}(n,n',\tilde{n},\tilde{n}')}{cR_\infty}\right) \approx \left(\frac{\delta_{l0}}{n^3} - \frac{\delta_{l'0}}{n'^3}\right) - \left(\frac{\delta_{\tilde{l}0}}{\tilde{n}^3} - \frac{\delta_{\tilde{l}'0}}{\tilde{n}'^3}\right)\frac{1/n^2 - 1/n'^2}{1/\tilde{n}^2 - 1/\tilde{n}'^2}, \quad (18)$$

in which $n', l' \to n, l$ is the transition to be predicted (for example, 2S–6P transition for $\nu_{2S-6P,\mathrm{SM}}$) and $\tilde{n}', \tilde{l}' \to \tilde{n}, \tilde{l}$ is the measured transition used as input (for example, 1S–2S transition for $\nu_{2S-6P,\mathrm{SM}}$). The sensitivity of $\nu_{2S-6P,\mathrm{SM}}$ is $\partial(\nu_{2S-6P,\mathrm{SM}}/cR_\infty)/\partial C = 0.134$. By contrast, using the 1S–3S transition instead of the 2S–6P transition leads to a 1.8-fold lower sensitivity ($\partial(\nu_{\mathrm{SM}}/cR_\infty)/\partial C = 0.074$) because the relative contribution of $C$ to the 1S–3S transition frequency is approximately twice as large as for the 2S–6P transition frequency. Combined with its 1.5-fold smaller uncertainty, $\nu_{2S-6P}$ therefore tests $C$ with 2.7-fold higher precision than the 1S–3S measurement[5]. Because the nuclear size correction $cR_\infty C_{\mathrm{NS}}\,r_\mathrm{p}^2/n^3$ scales as $1/n^3$ like the other bound-state QED corrections $cR_\infty C/n^3$, comparing $r_\mathrm{p}$ found from atomic hydrogen and from muonic hydrogen is a test of bound-state QED, as any missing or miscalculated terms in atomic hydrogen of the form $cR_\infty C/n^3$ would lead to a discrepancy. Furthermore, the precision with which $r_\mathrm{p}$ can be extracted from a given combination of measurements is therefore a direct measure of the precision of the implied QED test. Note that, because Extended Data Table 1 lists the corrections of $\nu_{2S-6P,\mathrm{SM}}$, which are approximately $-cR_\infty C/2^3$, the sensitivity relative to the listed corrections is $-8\times0.134 = -1.072$. For example, artificially removing the hadronic vacuum polarization shifts $\nu_{2S-6P,\mathrm{SM}}$ by $-1.072\times425.1$ Hz $= -0.46$ kHz, or approximately one experimental $\sigma$ of $\nu_{2S-6P}$.

We may also extract the 1S Lamb shift $\mathcal{L}(1S)$ by writing equation (1) as

$$E_{nlj} = chR_\infty f_{nlj}^{\mathrm{Dirac}} + \delta_{0l}\frac{h\mathcal{L}(1S)}{n^3} + h\delta\mathcal{L}(nlj), \quad (19)$$

in which $\mathcal{L}(1S) \equiv \mathcal{L}(1,0,1/2) = cR_\infty(f_{1,0,1/2}^{\mathrm{QED}} + C_{\mathrm{NS}}r_\mathrm{p}^2)$ and $\delta\mathcal{L}(nlj) = \mathcal{L}(nlj) - \delta_{0l}\mathcal{L}(1S)/n^3$ is the state-specific Lamb shift. By combining equation (19) for the 2S–6P and 1S–2S transitions and using $\nu_{2S-6P}$ and $\nu_{1S-2S}$ as inputs, we find $\mathcal{L}_{\mathrm{exp}}(1S) = 8{,}172{,}744.13(14)_{\mathrm{QED}}(3.56)_{\mathrm{exp}}$ kHz $= 8{,}172{,}744.1(3.6)$ kHz. Terms proportional to state-specific Lamb shifts contribute 224 MHz to $\mathcal{L}_{\mathrm{exp}}(1S)$. The uncertainty is dominated by the uncertainty of $\nu_{2S-6P}$ ('exp'), with the QED uncertainty of the state-specific Lamb shifts ('QED') being much smaller and contributions from $\nu_{1S-2S}$ (22 Hz) and physical constants (10 Hz from $\alpha$; $m_\mathrm{p}/m_\mathrm{e}$, $r_\mathrm{p}$, $R_\infty$ less than 0.1 Hz) negligible. As expected from the discussion above, $\mathcal{L}_{\mathrm{exp}}(1S)$ is 2.7-fold more precise than the next best determination using the 1S–3S measurement[5].

The corresponding QED prediction of the 1S Lamb shift is found by combining equation (1) with the muonic value of $r_\mathrm{p}$, giving $\mathcal{L}_{\mathrm{QED}}(1S) = 8{,}172{,}744.1(1.3)_{\mathrm{QED}}(1.0)_{r_\mathrm{p}}$ kHz $= 8{,}172{,}744.1(1.7)$ kHz. Its uncertainty is dominated by both QED uncertainty and the uncertainty of $r_\mathrm{p}$, whereas other sources are negligible (3 Hz from $\alpha$; $m_\mathrm{p}/m_\mathrm{e}$, $R_\infty$ less than 0.01 Hz). $\mathcal{L}_{\mathrm{exp}}(1S)$ and $\mathcal{L}_{\mathrm{QED}}(1S)$ are uncorrelated, as their respective QED uncertainties are uncorrelated[1]. They are in excellent agreement and test the 1S Lamb shift and thereby bound-state QED corrections to 0.5 ppm. Complementary tests of bound-state QED in strong electromagnetic fields with highly charged ions at present achieve a relative precision of $1\times10^{-4}$ but can be more sensitive to terms of high order in

$(Z\alpha)$ ($Z$ is the nuclear charge number; omitted elsewhere here because $Z = 1$ for atomic hydrogen)[47,67,68].

Finally, the QED prediction $\Delta\nu_{\mathrm{FS,QED}}(6P)$ for the 6P fine-structure splitting between the $6P_{1/2}^{F=1}$ and $6P_{3/2}^{F=1}$ levels is (using equations (14) and (15))

$$\begin{aligned}
\Delta\nu_{\mathrm{FS,QED}}(6P) &= E_{6,1,3/2}/h - E_{6,1,1/2}/h + \Delta\nu_{\mathrm{HFS}}(6P_{3/2}^{F=1}) \\
&\quad -\Delta\nu_{\mathrm{HFS}}(6P_{1/2}^{F=1}) \\
&= 405{,}164.51(1)\ \text{kHz.}
\end{aligned} \tag{20}$$

The nuclear size corrections in $\Delta\nu_{\mathrm{FS,QED}}(6P)$ are of order $\alpha^2 \times \alpha^4$ and $\alpha^2 \times \alpha^5$ and amount to 70 mHz, as the leading-order correction term $cR_\infty C_{\mathrm{NS}}r_\mathrm{p}^2 \propto \alpha^2 \times \alpha^2$ of equation (1) and corrections of order $\alpha^2 \times \alpha^3$ only apply to S-states[1].

## dc-Stark shift

Static stray electric fields in the 2S–6P spectroscopy region can lead to a dc-Stark shift of the observed transition frequency. Here the dc-Stark shift $\Delta\nu_{\mathrm{dc}}$ is well described as quadratic in strength $E = |\mathbf{E}|$ of the electric field $\mathbf{E}$ in all relevant experimental regimes, that is,

$$\Delta\nu_{\mathrm{dc}} = \beta_{\mathrm{dc}} E^2, \tag{21}$$

in which $\beta_{\mathrm{dc}}$ is the applicable quadratic dc-Stark shift coefficient. We distinguish two experimentally relevant field strength regimes: the stray-field regime ($E < 1\ \text{V m}^{-1}$), which covers the range of stray electric fields present in the experiment, and the bias-field regime ($E = 10\text{–}45\ \text{V m}^{-1}$), which covers the range of applied bias fields used to determine the stray electric fields. Notably, although equation (21) is found to approximately hold in either regime, the coefficient $\beta_{\mathrm{dc}}$ may differ, as is the case for the 2S–6P$_{3/2}$ transition (Section 1.4 in the Supplementary Methods). The quadratic behaviour arises because the energy levels contributing to the net shift are well separated in energy from the perturbed level in either of our regimes. However, the shift of the involved levels between the regimes can lead to substantially different energy separations and thereby different values of $\beta_{\mathrm{dc}}$.

The Faraday cage surrounding the spectroscopy region (Fig. 2b) shields it from external electric fields, including those used to draw the photoelectrons to the channel electron multipliers (whose input surfaces are held at 270 V; see Section 4.6.2 of ref. 23). The colloidal graphite coating on all surfaces of the Faraday cage suppresses stray electric fields from the surfaces themselves (from charged oxide layers, contact potentials from dissimilar conductors or local changes in the work function) by, ideally, forming a uniform conductive layer with a uniform work function (see Section 4.6.1 of ref. 23). However, effects such as imperfect shielding of external fields, imperfect graphite coating or temperature gradients leading to thermoelectric voltages or gradients in the work function can prevent the complete suppression of stray fields.

To address this, we measure the stray electric field by applying voltages to the six electrodes forming the Faraday cage (meshes at the top and bottom and four equal-sized segments of the cylinder wall) and using the atoms themselves as field sensors (see Section 4.6.7 of ref. 23), similar to approaches in refs. 5,69. Equal and opposite bias voltages are applied to opposing electrodes to create a bias electric field $\mathbf{E} = E_i\hat{\mathbf{i}}$ with strength $|E_i| = 10\text{–}45\ \text{V m}^{-1}$ along the given direction ($i = x, y, z$, as defined in Fig. 2b). By measuring the shifted 2S–6P transition frequency $\nu_\mathrm{e}(E_i)$ from a fit to the fluorescence line shape of several line scans with opposite-polarity values of $E_i$ and using the quadratic dependence $\nu_\mathrm{e}(E_i) = \beta_{\mathrm{dc},i}(E_i - \Delta E_i)^2 + \nu_\mathrm{e}(E_i = 0\ \text{V m}^{-1})$, we determine the stray electric field component $\Delta E_i$ along the given direction. This measurement also yields experimental values of $\beta_{\mathrm{dc},i}$ at the bias field strengths for each transition. Extended Data Fig. 3 shows examples of such stray field

measurements for both the 2S–6P$_{1/2}$ and 2S–6P$_{3/2}$ transitions, along with simulation results (see Section 1.4 in the Supplementary Methods). On average, each measurement includes eight line scans with non-zero bias field and there are 98 and 21 measurements for the 2S–6P$_{1/2}$ and 2S–6P$_{3/2}$ transitions, respectively. The stray electric field components determined during the three measurement runs are shown in Extended Data Fig. 4, in which we only include measurements using the 2S–6P$_{1/2}$ transition because using the 2S–6P$_{3/2}$ transition gives compatible, but substantially larger uncertainty, values of $\Delta E_i$. Overall, the stray electric field has a strength less than $1\ \text{V m}^{-1}$ and predominantly points along the axis of the detector cylinder (the $y$-direction).

Using these stray field measurements, we estimate the dc-Stark shift correction and its uncertainty for the transition frequencies. The determination of the relevant quadratic dc-Stark shift coefficients in the stray-field regime by the use of experimentally verified simulations is described in Section 1.4 in the Supplementary Methods. The weighted mean and standard deviation of each stray electric field component $\Delta E_i$ are determined over each of the three measurement runs, using the 2S–6P$_{1/2}$ stray field measurements, as shown in Extended Data Fig. 4. We treat each measurement run separately to account for differences in the detector assembly (the meshes in the detector cylinder were replaced between run A and run B; see Section 4.6.1 of ref. 23) and the month-long breaks in between the runs. Furthermore, we opt to use the standard deviation of the stray field components to estimate the dc-Stark shift uncertainty because we believe at least part of the variation of the stray fields to be physical in origin (as opposed to unaccounted excess measurement scatter) but are not confident that the resulting variations in the dc-Stark shift will average out. With this, we find dc-Stark shifts $\Delta\nu_{\mathrm{dc}}$ of 0.20(21) kHz and −0.02(6) kHz for the 2S–6P$_{1/2}$ and 2S–6P$_{3/2}$ transitions, respectively, in which the uncertainty is dominated by the stray electric fields in the first case and by coefficients in the second case, and shifts for both transitions mainly determined by the dominant stray electric field component along the $y$-direction. $\nu_{1/2}$ and $\nu_{3/2}$ have been corrected for the dc-Stark shift by subtracting the corresponding value. The Pearson correlation coefficient of the correction between the 2S–6P$_{1/2}$ and 2S–6P$_{3/2}$ transitions is found to be $r = -0.30$ by propagation of uncertainty.

## Data availability

The experimental data and simulation results that support the findings of this study are available from Zenodo[70].

## Code availability

All code used for the data analysis and simulations of this study is available from the corresponding author.

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

**Acknowledgements** We thank S. G. Karshenboim for helpful discussions and W. Simon, K. Linner and H. Brückner for technical support. Computations were performed on the HPC systems at the Max Planck Computing and Data Facility. This work was supported by European Research Council (ERC) grant H-SPECTR (grant agreement ID 101141942). L.M. acknowledges a Feodor Lynen Fellowship from the Alexander von Humboldt Foundation. V.W. acknowledges support from the International Max Planck Research School for Advanced Photon Science (IMPRS-APS). A.M. and A.G. acknowledge support from the German Research Foundation (DFG) (project IDs 390524307 and EXC-2111-390814868, respectively). R.P. acknowledges support from the PRISMA+ Cluster of Excellence (EXC 2118/1) funded by the DFG in the German Excellence Strategy (project ID 39083149). T.W.H. acknowledges support from the Carl Friedrich von Siemens Foundation and the Max Planck Foundation.

**Author contributions** L.M. and V.W. set up, maintained and performed the experiment. L.M. wrote the software for and set up the data acquisition system. L.M. conducted the data analysis, which V.W. independently reproduced. A.M. and L.M. performed modelling and simulations. T.U., L.M. and V.W. evaluated the QED predictions and tests. L.M. prepared the manuscript. All authors (L.M., V.W., A.M., A.G., R.P., T.W.H. and T.U.) contributed to the discussion and analysis of the systematic uncertainties and edited the manuscript.

**Funding** Open access funding provided by Max Planck Society.

**Competing interests** The authors declare no competing interests.

**Additional information**
**Correspondence and requests for materials** should be addressed to Lothar Maisenbacher.

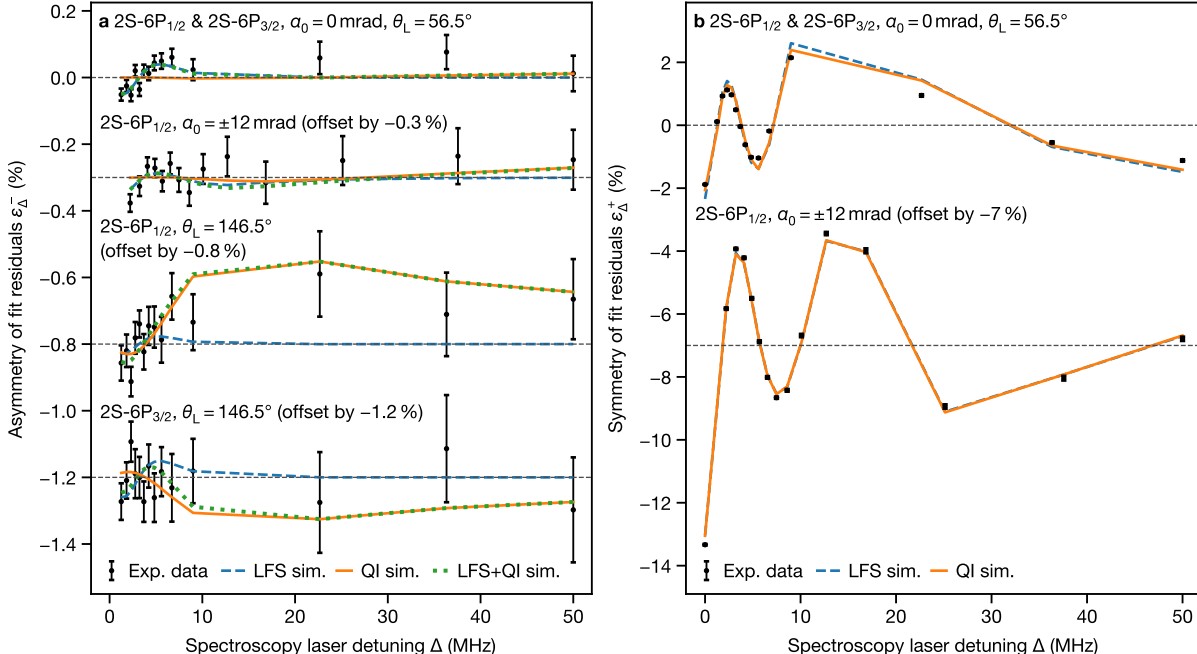

**Extended Data Fig. 1 | Test of line shape models. a**, Detuning asymmetry of fit residuals $\varepsilon_\Delta^-$ at spectroscopy laser detuning $\Delta$ for four different experimental settings, with experimental data (black circles), LFS simulation (dashed blue line) and QI simulation (solid orange line) shown. The sum of $\varepsilon_\Delta^-$ for LFS and QI simulations is also shown (dotted green line). The normalized fit residuals $\varepsilon_\Delta$ are given by $(y_\Delta - y_{\mathrm{fit},\Delta})/y_{\mathrm{fit},\Delta}$, in which $y_\Delta$ is the experimental or simulated signal and $y_{\mathrm{fit},\Delta}$ the corresponding value from the Voigt or Voigt doublet line shape fit to the signal, and their detuning asymmetry and symmetry are defined as $\varepsilon_\Delta^- = (\varepsilon_\Delta - \varepsilon_{-\Delta})/2$ and $\varepsilon_\Delta^+ = (\varepsilon_\Delta + \varepsilon_{-\Delta})/2$. Note that the curves have been offset (dashed grey horizontal lines) to enhance visibility. Top, for data taken at zero atomic beam offset angle $\alpha_0$ and at magic polarization angle $\theta_L = 56.5°$, in which the QI effect is strongly suppressed, the experimentally observed asymmetry $\varepsilon_\Delta^-$ is well explained by the LFS simulation. All relevant data from measurement run B are used. See Fig. 2c for an example line shape. Second from top, for $\alpha_0 = \pm 12$ mrad (and magic polarization angle; see Fig. 3c for an example line

shape), the LFS simulation predicts a different asymmetry compared with zero $\alpha_0$, which again matches the experiment well. This difference in asymmetry is directly observed in the transition frequency as shown in Fig. 3d. Third from top, for a non-magic polarization angle of $\theta_L = 146.5°$, the QI effect leads to a strong asymmetry at large detunings $\Delta \gtrsim 5$ MHz $\approx \Gamma$, which is observed in the experimental data (shown for the 2S–6P$_{1/2}$ transition; $\alpha_0 = 0$ mrad). At smaller detunings, both the QI effect and LFS contribute to the asymmetry and the sum of their asymmetries describes the experimental data well. Bottom, for the non-magic polarization angle for the 2S–6P$_{3/2}$ transition, the asymmetry from the QI effect has the opposite sign and is about twofold smaller compared with the 2S–6P$_{1/2}$ transition. **b**, Detuning symmetry of fit residuals $\varepsilon_\Delta^+$ for the first two cases of panel **a**. The deviations are caused by non-Gaussian broadening and saturation effects, which are not included in the fitted line shape models but are included in the simulations. Only results for the top detector are shown, as the detector signals are highly correlated.

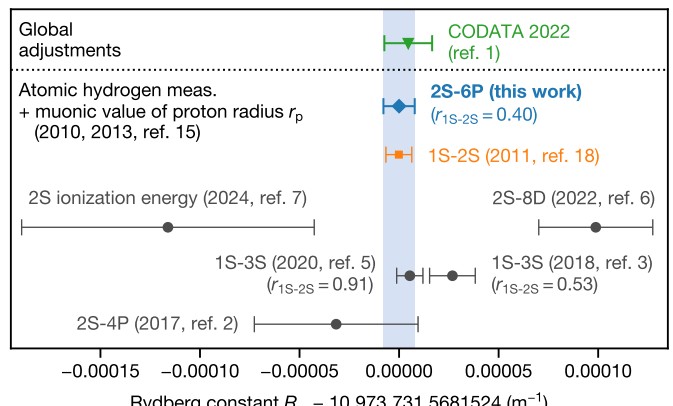

**Extended Data Fig. 2 | Rydberg constant $R_\infty$ from atomic hydrogen.** The $R_\infty$ values are determined by combining each atomic hydrogen measurement[2,3,5-7,18] with the proton radius $r_p$ from muonic hydrogen spectroscopy[8,9,15] and equation (1). The large, correlated QED uncertainty of equation (1) for S-states (along with the common $r_p$ input) leads to a large correlation between some $R_\infty$ values, reducing their sensitivity as QED tests ($r_{1S-2S}$ is the Pearson correlation coefficient with $R_\infty$ from 1S–2S measurement[18]; $r_{1S-2S} < 0.1$ if not shown). In particular, $r_{1S-2S} = 0.91$ for the 2020 1S–3S measurement because of the dominant 1S level QED uncertainty[5], whereas $r_{1S-2S} = 0.40$ for the 2S–6P measurement because of the (eightfold) lower QED uncertainty of the 2S level. For reference, we show the value from the CODATA 2022 global adjustment[1], which includes an uncertainty expansion factor to account for the scatter of the data. The addition of the high-precision, relatively uncorrelated 2S–6P measurement is likely to greatly reduce the effective scatter of future global adjustments, as less precise measurements are removed from the adjustment. Error bars show one-standard-deviation uncertainties.

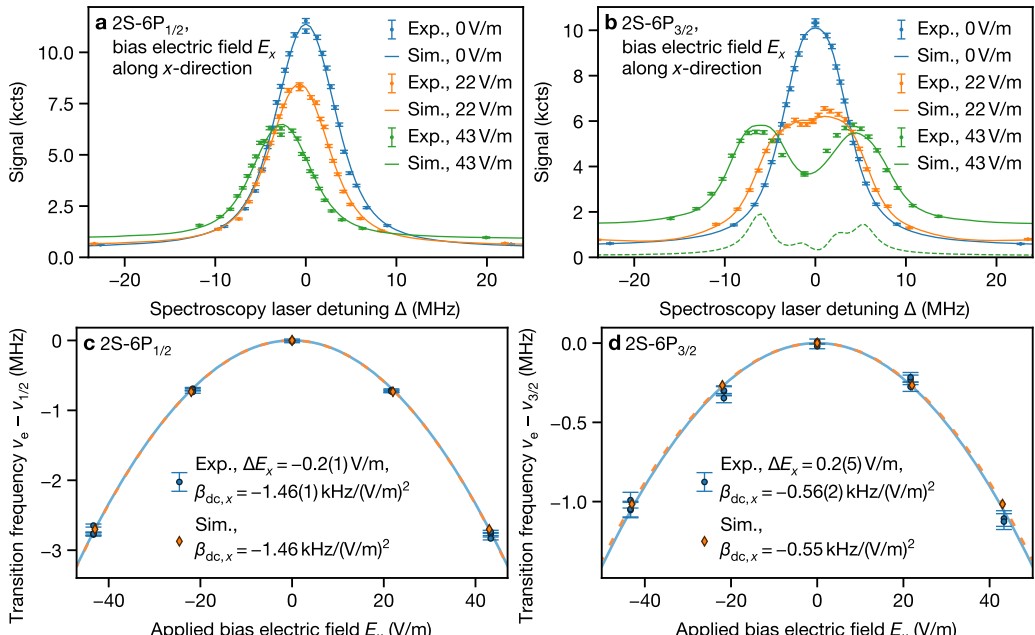

**Extended Data Fig. 3 | In situ determination of stray electric fields. a**, The line of the 2S–6P$_{1/2}$ transition (circles) shifts as a bias electric field $E_x$ (here along the $x$-direction; Fig. 2b) is applied. Velocity group $\tau_9$ is shown, as its mean speed is closest to the mean speed $\langle \bar{v} \rangle$ of all velocity groups. Solid lines show the simulated line shape for each value of $E_x$ (scaled to match experimental data). A Voigt line shape fit (not shown) is used to find the resonance frequency. **b**, The line shape of the 2S–6P$_{3/2}$ transition (circles) splits into two distinct components as $E_x$ is increased and excitations of 6S and 6D levels become allowed. The simulated line shapes (solid lines) reproduce this splitting well and describe the relative amplitudes of the resulting components reasonably well. A simulation of a single atomic trajectory without Doppler broadening (dashed green line; scaled for visibility; $E_x = 43$ V m$^{-1}$) reveals the more complex underlying substructure. A Voigt doublet line shape fit (not shown) is used to find the centre-of-mass resonance frequency of the line shape. **c**, A single stray field measurement using the 2S–6P$_{1/2}$ transition. The Doppler-free transition frequency $\nu_e(E_x)$ (blue circles; error bars show statistical uncertainty only), found by the usual Doppler shift extrapolation, varies quadratically with $E_x$. A quadratic fit (solid blue line) gives the stray electric field component $\Delta E_x$ (Extended Data Fig. 4) and quadratic dc-Stark shift coefficient $\beta_{dc}$. The simulation (orange diamonds) and its fit (dashed orange line) agree with the experiment. The field-induced shift of the transition frequencies coincides with the energy shift of the excited level of the 2S–6P$_{1/2}$ transition. **d**, Similar to panel **c** but using the 2S–6P$_{3/2}$ transition, again showing good agreement between experiment and simulation. The shift of the centre-of-mass transition frequency is much smaller than both the shift of either the resolved line shape components of panel **b** or the energy shift of the excited level of the 2S–6P$_{3/2}$ transition. Only results for the top detector are shown, as the detector signals are highly correlated. See Methods and Supplementary Methods for details.

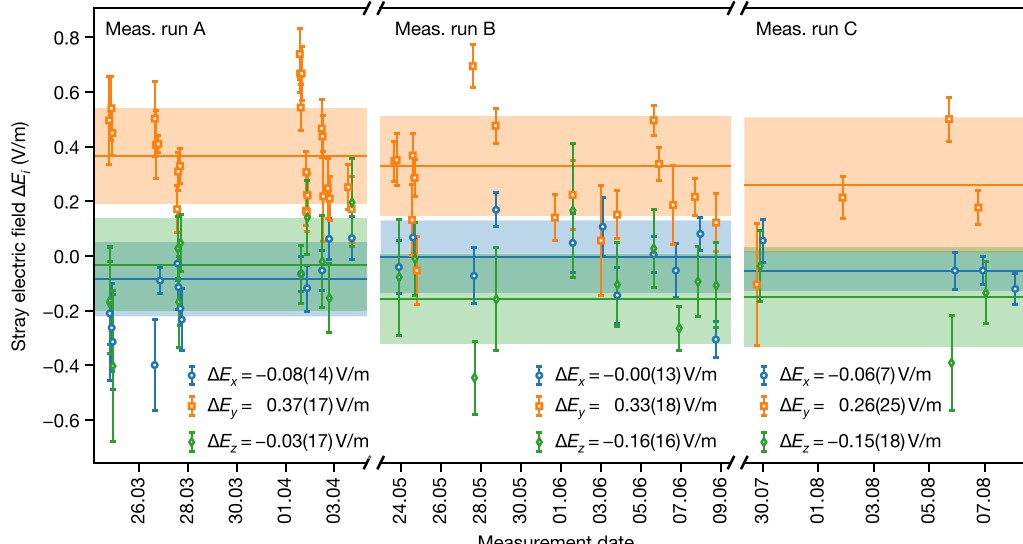

**Extended Data Fig. 4 | Stray electric fields during measurement runs.**
The stray electric field components $\Delta E_x$ (blue circles), $\Delta E_y$ (orange squares) and $\Delta E_z$ (green diamonds) in the 2S–6P spectroscopy region, as determined by spectroscopy of the 2S–6P$_{1/2}$ transition with applied bias electric fields (Extended Data Fig. 3 and Methods). See Fig. 2b for the coordinate system definition. Error bars show statistical uncertainties of the detector-averaged data. Shaded areas show the standard deviation for each component and measurement run, with weighted means and standard deviations (in parentheses) shown in the legends. The stray electric field predominantly points along the $y$-direction, which is the axis of the detector cylinder.

**Extended Data Table 1 | Contributions to the SM prediction $v_{2S-6P,SM}$ of the 2S–6P fine-structure centroid**

| Contribution | Value (Hz) | QED-only uncertainty (Hz) |
|---|---:|---:|
| Dirac eigenvalue ($cR_\infty f^{\text{Dirac}}$) | 730,691,293,379,477.0 | — |
| Lamb shift $\mathcal{L}$(2S-6P) | −1,044,768,691.8 | 178.6 |
| Relativistic recoil | −340,939.5 | 0.1 |
| One-photon corrections ($\propto \alpha^2 \times \alpha^3$) | | |
| Self-energy | −1,071,052,910.8 | 0.8 |
| Vacuum polarization | 26,853,096.2 | 0.0 |
| Muonic ($\mu^+\mu^-$) vacuum pol. | 633.5 | 0.0 |
| Hadronic vacuum pol. | 425.1 | 10.2 |
| Two-photon corrections ($\propto \alpha^2 \times \alpha^4$) | −91,497.4 | 126.4 |
| Three-photon corrections ($\propto \alpha^2 \times \alpha^5$) | −214.6 | 45.8 |
| Finite nuclear size and polarizability | | |
| $cR_\infty C_{\text{NS}} r_p^2 \propto \alpha^2 \times \alpha^2$ | −138,130.9 | 0.0 |
| $\propto \alpha^2 \times \alpha^3$ | 13.6 | 1.7 |
| $\propto \alpha^2 \times \alpha^4$ | −123.5 | 52.8 |
| $\propto \alpha^2 \times \alpha^5$ | 0.5 | 0.2 |
| Radiative-recoil corrections | 1540.2 | 102.2 |
| Nucleus self-energy | −584.2 | 21.6 |
| Total | 730,690,248,610,785.2 | 178.6 |

The naming and grouping of the contributions follow ref. 1. Indented entries detail subcontributions to the Lamb shift (that is, the sum of all QED corrections). The values are calculated by combining equation (1) with the measurement of the 1S–2S transition frequency[18] and the muonic value of $r_p$ (ref. 15). $\alpha^2 \times \alpha^n$ indicates the leading order in $\alpha$ of selected contributions, with the first $\alpha^2$ factor absorbed in $R_\infty$ in equation (1). The QED-only uncertainty is the uncertainty excluding contributions from $r_p$, $\alpha$, $m_p/m_e$ and $v_{1S-2S}$. See Methods for details.

**Extended Data Table 2 | Definition and properties of atomic velocity groups $\tau_i$**

| Velocity group | $\tau$ (µs) | $P_{2S}$ | $\bar{v}$ (m/s) | $\Delta v_t$ (m/s) | $\Gamma_F$ (MHz) |
|---|---|---|---|---|---|
| $\tau_1$ | 10...60 | $2.1 \times 10^{-2}$ | 253(5) | 3.32(6) | 9.6(4) |
| $\tau_2$ | 60...110 | $2.1 \times 10^{-2}$ | 252(5) | 3.32(6) | 9.6(4) |
| $\tau_3$ | 110...160 | $2.0 \times 10^{-2}$ | 250(5) | 3.29(6) | 9.5(4) |
| $\tau_4$ | 160...210 | $1.9 \times 10^{-2}$ | 247(5) | 3.23(5) | 9.4(4) |
| $\tau_5$ | 210...260 | $1.7 \times 10^{-2}$ | 242(4) | 3.14(5) | 9.2(4) |
| $\tau_6$ | 260...310 | $1.5 \times 10^{-2}$ | 236(4) | 2.99(4) | 8.9(4) |
| $\tau_7$ | 310...360 | $1.3 \times 10^{-2}$ | 227(4) | 2.79(4) | 8.5(4) |
| $\tau_8$ | 360...410 | $1.2 \times 10^{-2}$ | 217(3) | 2.59(3) | 8.1(4) |
| $\tau_9$ | 410...510 | $9.9 \times 10^{-3}$ | 202(3) | 2.32(2) | 7.7(4) |
| $\tau_{10}$ | 510...610 | $8.9 \times 10^{-3}$ | 183(2) | 2.01(2) | 7.1(4) |
| $\tau_{11}$ | 610...710 | $8.6 \times 10^{-3}$ | 165(2) | 1.76(1) | 6.7(3) |
| $\tau_{12}$ | 710...910 | $7.1 \times 10^{-3}$ | 145(2) | 1.50(1) | 6.2(3) |
| $\tau_{13}$ | 910...1210 | $6.4 \times 10^{-3}$ | 120(1) | 1.19(1) | 5.8(3) |
| $\tau_{14}$ | 1210...1510 | $7.0 \times 10^{-3}$ | 98(1) | 0.94(1) | 5.5(4) |
| $\tau_{15}$ | 1510...2010 | $6.0 \times 10^{-3}$ | 81(1) | 0.76(2) | 5.5(5) |
| $\tau_{16}$ | 2010...2560 | $6.4 \times 10^{-3}$ | 65(1) | 0.58(2) | 5.5(6) |

Each velocity group integrates the fluorescence signal over a range of delay times $\tau$, with $\tau=0$ µs set by the blocking of the 1S–2S preparation laser. Some properties are found by numerically modelling the atomic beam and its interaction with the 1S–2S preparation laser and the 2S–6P spectroscopy laser (see Methods and Section 1.1 in the Supplementary Methods): $P_{2S}$ is the average excitation probability to the metastable 2S level; $\bar{v}$ is the mean speed of atoms contributing to the fluorescence signal; $\Delta v_t$ is the FWHM of the transverse velocity distribution of 2S atoms. The modelling has been experimentally verified by comparison with the signal of each velocity group. $\Gamma_F$ is the experimentally observed FWHM linewidth of the 2S–6P transition. All numbers are weighted averages over the data groups given in Extended Data Table 3 (with the weights $w_{2S–6P}$ given therein); numbers in parentheses are weighted standard deviations over the data groups.

## Extended Data Table 3 | Experimental parameters of data groups

| Data group | Meas. run | Transition | $\alpha_0$ (mrad) | $\theta_L$ (°) | $P_{2S-6P}$ (µW) | FCs | Line scans per detector Top | Line scans per detector Bot. | $v_{cut\text{-}off}$ (m/s) | $P_{1S-2S}$ (W) | $w_{2S-6P}$ (%) |
|---|---|---|---|---|---|---|---|---|---|---|---|
| G1A | A |  |  |  | 30 | 13 | 285 | 285 | $31^{+56}_{-33}$ | $1.30^{+0.50}_{-0.40}$ | 5.7 |
| G1B | B |  |  |  | 30 | 18 | 148 | 141 | $65^{+39}_{-44}$ | $1.15^{+0.15}_{-0.15}$ | 4.6 |
| G1C | C | $2S\text{-}6P_{1/2}$ | 0.0 | 56.5 | 30 | 16 | 77 | 68 | $52^{+32}_{-18}$ | $1.05^{+0.10}_{-0.10}$ | 2.4 |
| G2 | B |  |  |  | 20 | 18 | 147 | 138 | $65^{+46}_{-41}$ | $1.15^{+0.15}_{-0.15}$ | 3.5 |
| G3 | B |  |  |  | 10 | 18 | 598 | 564 | $66^{+43}_{-43}$ | $1.15^{+0.15}_{-0.15}$ | 8.8 |
| G4 | B |  |  |  | 30 | 3 | 34 | 31 | $41^{+26}_{-23}$ | $1.10^{+0.05}_{-0.05}$ | 0.6 |
| G5 | B | $2S\text{-}6P_{1/2}$ | 0.0 | 146.5 | 20 | 3 | 34 | 31 | $42^{+28}_{-27}$ | $1.10^{+0.05}_{0.00}$ | 0.9 |
| G6 | B |  |  |  | 10 | 3 | 132 | 119 | $42^{+34}_{-29}$ | $1.10^{+0.05}_{-0.05}$ | 3.0 |
| G7A | A |  |  |  | 15 | 3 | 162 | 162 | $32^{+34}_{-28}$ | $1.25^{+0.10}_{-0.30}$ | 8.3 |
| G7B | B | $2S\text{-}6P_{3/2}$ | 0.0 | 56.5 | 15 | 17 | 143 | 116 | $57^{+42}_{-43}$ | $1.05^{+0.15}_{-0.10}$ | 13.4 |
| G8 | B |  |  |  | 10 | 18 | 151 | 124 | $56^{+39}_{-44}$ | $1.05^{+0.15}_{-0.10}$ | 9.2 |
| G9 | B |  |  |  | 5 | 18 | 568 | 461 | $57^{+43}_{-49}$ | $1.05^{+0.15}_{-0.10}$ | 25.7 |
| G10 | B |  |  |  | 15 | 2 | 21 | 20 | $42^{+23}_{-19}$ | $1.10^{+0.05}_{-0.05}$ | 2.4 |
| G11 | B | $2S\text{-}6P_{3/2}$ | 0.0 | 146.5 | 10 | 2 | 22 | 21 | $41^{+26}_{-19}$ | $1.10^{+0.05}_{-0.05}$ | 1.7 |
| G12 | B |  |  |  | 5 | 2 | 87 | 80 | $36^{+35}_{-23}$ | $1.10^{+0.05}_{-0.05}$ | 5.9 |
| G13 | C | $2S\text{-}6P_{1/2}$ | ±7.5 | 56.5 | 30 | 3 | 106 | 106 | $61^{+33}_{-22}$ | $1.10^{+0.05}_{-0.05}$ | 0.8 |
| G14 | C |  | ±12.0 |  | 30 | 11 | 440 | 394 | $63^{+38}_{-43}$ | $1.05^{+0.10}_{-0.15}$ | 2.9 |

Experimental data taken in the same measurement run, and for the same transition, atomic beam offset angle $\alpha_0$, laser polarization angle $\theta_L$ and spectroscopy laser power $P_{2S-6P}$ are grouped, forming the 17 data groups listed here. Mean values of cut-off speed $v_{cut\text{-}off}$ and preparation laser power $P_{1S-2S}$ are given, along with the differences to the maximum (superscript) and minimum (subscript) values. Note that approximate values of $v_{cut\text{-}off}$ were used to determine simulation corrections (Extended Data Table 5). FCs, number of freezing cycles contained in the group; line scans per detector, number of line scans contained in the group for top and bottom (bot.) detectors; $w_{2S-6P}$, statistical weight in the 2S–6P fine-structure centroid $v_{2S-6P}$. See Methods for details.

**Extended Data Table 4 | Corrections Δν and uncertainties σ for the determination of the 2S–6P$_{1/2}$ ($\nu_{1/2}$) and the 2S–6P$_{3/2}$ ($\nu_{3/2}$) transition frequencies and the 6P fine-structure splitting Δν$_{FS}$(6P)**

| Contribution | $\nu_{1/2}$ | | $\nu_{3/2}$ | | $\Delta\nu_{FS}(6P)$ | |
|---|---|---|---|---|---|---|
| | Δν (kHz) | σ (kHz) | Δν (kHz) | σ (kHz) | Δν (kHz) | σ (kHz) |
| First-order Doppler shift | −0.32 | 0.49 | 0.67 | 0.60 | 1.00 | 0.77 |
| Extrapolation (statistical) | −0.32 | 0.49 | 0.67 | 0.60 | 1.00 | 0.77 |
| Simulation of atom speeds | — | 0.02 | — | 0.01 | — | 0.02 |
| Simulation corrections | 0.88 | 0.43 | 1.14 | 0.22 | 0.26 | 0.53 |
| Light force shift | 0.78 | 0.23 | 1.33 | 0.16 | 0.54 | 0.14 |
| Quantum interference shift | 0.25 | 0.36 | −0.05 | 0.15 | −0.29 | 0.51 |
| Second-order Doppler shift | −0.15 | 0.01 | −0.14 | 0.01 | 0.00 | 0.00 |
| dc-Stark shift | 0.20 | 0.21 | −0.02 | 0.06 | −0.22 | 0.23 |
| BBR-induced shift | 0.28 | 0.01 | 0.28 | 0.01 | 0.00 | 0.00 |
| Zeeman shift | 0.00 | 0.02 | 0.00 | 0.11 | 0.00 | 0.09 |
| Pressure shift | 0.00 | 0.02 | 0.00 | 0.02 | 0.00 | 0.00 |
| Sampling bias | 0.00 | 0.04 | 0.00 | 0.08 | 0.00 | 0.04 |
| Signal background | 0.00 | 0.03 | 0.00 | 0.04 | 0.00 | 0.05 |
| Laser spectrum | 0.00 | 0.07 | 0.00 | 0.07 | 0.00 | 0.00 |
| Frequency standard | 0.02 | 0.01 | 0.01 | 0.01 | 0.00 | 0.00 |
| Subtotal (experiment-specific contributions) | 1.06 | 0.69 | 2.08 | 0.66 | 1.04 | 0.97 |
| Recoil shift | −1176.03 | 0.00 | −1176.03 | 0.00 | 0.00 | 0.00 |
| Total (all corrections) | −1174.97 | 0.69 | −1173.95 | 0.66 | 1.04 | 0.97 |

All uncertainties correspond to one standard deviation. Indented entries detail subcontributions to the first-order Doppler shift and simulation corrections. The sum of the subcontributions may differ from the given total owing to rounding. BBR, blackbody radiation.

**Extended Data Table 5 | Contributions to simulation correction uncertainties for the 2S–6P$_{1/2}$ ($v_{1/2}$) and 2S–6P$_{3/2}$ ($v_{3/2}$) transition frequencies and the 6P fine-structure centroid $v_{2S-6P}$ and splitting $\Delta v_{FS}$(6P)**

| Contribution | Variation | Sim. | $v_{1/2}$ $\sigma$ (Hz) | $v_{3/2}$ $\sigma$ (Hz) | $v_{2S\text{-}6P}$ $\sigma$ (Hz) | $v_{2S\text{-}6P}$ $r$ | $\Delta v_{FS}$(6P) $\sigma$ (Hz) | $\Delta v_{FS}$(6P) $r$ |
|---|---|---|---|---|---|---|---|---|
| Nozzle temperature $T_N$ | $\pm 100$ mK | Total | 84 | 25 | 33 | 0.01 | 77 | 0.42 |
| | | LFS | 80 | 28 | 34 | 0.08 | 74 | 0.38 |
| | | QI | 3 | 1 | 1 | −0.18 | 3 | 0.26 |
| | | SOD | 1 | 1 | 1 | 0.78 | 0 | 0.97 |
| Nozzle radius $r_1$ | $^{+0.1}_{-0.5}$ mm* | Total | 89 | 34 | 52 | 1.00 | 55 | 1.00 |
| | | LFS | 43 | 42 | 42 | 1.00 | 5 | 0.99 |
| | | QI | 34 | 19 | 2 | −0.99 | 53 | −1.00 |
| | | SOD | 11 | 11 | 11 | 1.00 | 0 | 1.00 |
| Cut-off speed $v_{cut\text{-}off}$ | min.–max.*,† | Total | 21 | 23 | 16 | −0.19 | 31 | −0.05 |
| | | LFS | 23 | 27 | 26 | 1.00 | 16 | 0.80 |
| | | QI | 16 | 6 | 3 | −0.79 | 19 | −0.40 |
| | | SOD | 7 | 7 | 7 | 1.00 | 0 | 1.00 |
| Atomic beam aperture width | $\pm 0.1$ mm | Total | 133 | 32 | 66 | 1.00 | 100 | 1.00 |
| | | LFS | 138 | 40 | 72 | 1.00 | 98 | 1.00 |
| | | QI | 4 | 7 | 6 | 1.00 | 7 | 0.29 |
| | | SOD | 1 | 0 | 0 | −1.00 | 1 | −1.00 |
| Atomic beam offset angle $\alpha_0$ | $\pm 1$ mrad | Total | 139 | 126 | 123 | 0.74 | 41 | 0.96 |
| | | LFS | 137 | 130 | 123 | 0.72 | 35 | 0.97 |
| | | QI | 4 | 3 | 1 | −0.83 | 7 | −1.00 |
| | | SOD | 0 | 0 | 0 | 0.90 | 0 | 0.78 |
| 1S-2S preparation laser power $P_{1S\text{-}2S}$ and detuning | min.–max.*,‡ | Total | 67 | 33 | 45 | 1.00 | 33 | 1.00 |
| | | LFS | 54 | 34 | 41 | 1.00 | 21 | 1.00 |
| | | QI | 8 | 4 | 4 | 0.07 | 12 | −1.00 |
| | | SOD | 5 | 4 | 4 | 0.84 | 2 | 0.89 |
| 2S-6P spectroscopy laser power $P_{2S\text{-}6P}$ | $\pm 10$ %* | Total | 59 | 53 | 55 | 1.00 | 7 | 1.00 |
| | | LFS | 35 | 62 | 53 | 1.00 | 28 | 1.00 |
| | | QI | 25 | 9 | 2 | −1.00 | 34 | −1.00 |
| | | SOD | 0 | 0 | 0 | 0.88 | 0 | 0.99 |
| 2S-6P spectroscopy laser polarization angle $\theta_L$ | $\pm 3°$ | Total | 355 | 152 | 19 | −1.00 | 507 | −1.00 |
| | | LFS | 0 | 0 | 0 | −1.00 | 0 | −1.00 |
| | | QI | 355 | 152 | 19 | −1.00 | 507 | −1.00 |
| | | SOD | 0 | 0 | 0 | −1.00 | 0 | −1.00 |
| Detection efficiency | see § | Total | 33 | 14 | 2 | −0.99 | 47 | −1.00 |
| | | LFS | 0 | 0 | 0 | 0.33 | 0 | 0.33 |
| | | QI | 33 | 14 | 2 | −0.99 | 47 | −1.00 |
| | | SOD | 0 | 0 | 0 | −1.00 | 0 | −1.00 |
| Monte Carlo uncertainty of trajectory simulation | 10 rep.‖ | Total | 28 | 16 | 14 | 0.00 | 32 | 0.00 |
| | | LFS | 28 | 16 | 14 | 0.00 | 32 | 0.00 |
| | | QI | 2 | 2 | 1 | 0.00 | 2 | 0.00 |
| | | SOD | 0 | 0 | 0 | 0.00 | 0 | 0.00 |
| All contributions | | Total | 434 | 216 | 170 | −0.31 | 532 | −0.26 |
| | | LFS | 227 | 164 | 169 | 0.67 | 137 | 0.80 |
| | | QI | 360 | 155 | 20 | −0.99 | 514 | −1.00 |
| | | SOD | 14 | 14 | 14 | 0.98 | 2 | 0.98 |

For each contribution, the simulations were repeated with corresponding input parameters adjusted by the given variation about their optimal value. The resulting one-standard-deviation uncertainties $\sigma$ for the light force (LFS), QI and second-order Doppler (SOD) shifts are given, along with their combined uncertainty. Pearson correlation coefficients $r$ between the uncertainties for $v_{1/2}$ and $v_{3/2}$ corresponding to the determined uncertainties for $v_{2S-6P}$ and $\Delta v_{FS}$(6P) are also given. The simulation correction uncertainties given in Table 1 and Extended Data Table 4 correspond to the uncertainties given here added in quadrature over the contributions, as reproduced in the last row. The effect on the mean speeds of the velocity groups and thereby the transition frequencies is excluded here and listed separately in Table 1 and Extended Data Table 4. See Section 1.1 in the Supplementary Methods for details. *A multiplicative factor of $1/\sqrt{3}$ is applied to convert the half-width of box-like variations to one-standard-deviation uncertainties. †Mean, minimum and maximum values of $v_{cut\text{-}off}$ were taken to be 30, −10 and 90 m s$^{-1}$ for measurement run A and 65, 0 and 130 m s$^{-1}$ for measurement runs B and C (see Extended Data Table 3 for the determined values of $v_{cut\text{-}off}$ for each data group). ‡Mean, minimum and maximum values of $P_{1S\text{-}2S}$ used for each data group are listed in Extended Data Table 3. The detuning for each value of $P_{1S\text{-}2S}$ was found by a linear regression of measured detunings as a function of $P_{1S\text{-}2S}$ in each data group. §Detection efficiency uncertainty is estimated by varying the assumed transparencies of the meshes inside the detector cylinder and repeating the Monte Carlo particle tracing simulation of the spatial detection efficiency (see Section 1.2 in the Supplementary Methods). ‖The standard deviation over ten repetitions with randomly drawn trajectories is used as an estimate of the uncertainty from the randomness of the Monte Carlo trajectory simulation.