## [Peer Review file · Nature]

Sub-part-per-trillion test of the Standard Model with atomic hydrogen

Corresponding Author: Dr Lothar Maisenbacher

Version 0:

Reviewer comments:

Referee #1

(Remarks to the Author)

The manuscript entitled “Sub-part-per-trillion test of the Standard Model with atomic hydrogen” presents results from a thorough and high-level experimental work.

This work concerns the measurement of the 2S-6P transition frequency in hydrogen atoms with a relative uncertainty of 6.6×10^{-13} , leading to the determination of the proton radius with a relative uncertainty of 1.8×10^{-3} , a bit more than 2.5 -fold compared to other determinations from hydrogen spectroscopy. This result confirms definitively the value obtained by muon spectroscopy and closes, in my opinion, definitively the debate on the famous proton puzzle. The remarkable agreement with the Standard Model (SM) prediction using the value of the proton radius (from the muon) constitutes a precise test of the SM and QED calculations, almost competitive with the test using the electron's $g-2$.

The experiment is described in great details. The exhaustive analysis of several noise sources and systematic biases presented in “The Method” and “The Supplementary Method” reflects both the in-depth and rigor of the experimental methodology. I have reread the manuscript several times and it is difficult to find any major criticisms, as this paper is written with remarkable clarity and precision.

I have only two suggestions to help clarify some systematic effects

1) The quantum interference effect and the concept of the magic polarization angle are briefly mentioned in the main text and refer only to two references that may be difficult for non-specialists to understand. Even though the evaluation of the related systematic effect is detailed in the Method and Supplementary Method, the document would gain in clarity if the authors took the time to provide an intuitive explanation of this magic polarization angle and its link to quantum interference (in few sentences).

2) In S2.8 Recoil shift, authors may add a sentence indicating why Gouy phase and wavefront curvature are neglected.

In conclusion, I strongly recommend the publication of this paper and congratulate the authors on their excellent work.

Referee #2

(Remarks to the Author)

In the manuscript “Sub-part-per-trillion test of the Standard Model with atomic hydrogen” by Maisenbacher et al., the authors determine two transition frequencies of hydrogen atoms contained in a cryogenic beam to a precision of 0.9 parts per trillion using single-photon laser spectroscopy. The first transition was excited between a hyperfine magnetic substate of the metastable 2S manifold with a total electronic angular momentum quantum number of $J=1/2$, total angular momentum quantum number $F=0$, and total angular momentum magnetic quantum number $m_F=0$ (denoted as $2S_{1/2}^{F=0}, m_F=0$); and a

substate of the 6P manifold with the quantum numbers $J=1/2$, $F=1$, and $m_F=0$ ($6S^{F=1}_{1/2}, m_F=0$). The second transition was excited between the substates $2S^{F=0}_{1/2}, m_F=0$ and $6S^{F=1}_{3/2}, m_F=0$. From this the authors calculated the so-called 2S-6P transition frequency between the hyperfine centroid of the 2S manifold and the fine-structure centroid of the 6P manifold as 730690248610.79(48) kHz. By comparing the results with quantum electrodynamics calculations and the 1s-2s transition frequency measured by the same group in a previous publication, the root-mean-square charge radius of the proton was determined as 0.8406(15) fm. This result is 2.5 times more precise than any previous determination based on spectroscopy of hydrogen while also coinciding with the charge radius determined by laser spectroscopy of the 2S-2P transition of muonic hydrogen atoms. The value significantly disagrees with the CODATA 2014 value of the proton charge radius.

I recommend publication of this work in Nature due to its central importance in testing the validity of quantum electrodynamics (QED) which forms the basis of the standard model of particle physics and specifically in resolving the so-called "proton radius puzzle". The spatial distribution of electric charge within a proton is affected by the complex dynamics of the valence and virtual sea quarks and of the gluons that mediate the strong interaction. The proton charge radius cannot yet be easily calculated from first principles using quantum chromodynamics (QCD) because of the non-perturbative nature of the strong interaction at the low momentum transfer regions relevant to the inner structure of the proton. So far, the only ab initio way of calculating the electric and magnetic radii has been provided by lattice QCD methods containing numerous simplifying approximations. Prior to 2010, the experimental values of the charge radius were determined either by scattering high energy beams of electrons on proton targets and carrying out a phenomenological analysis to extrapolate the scattering data to zero momentum transfer, or by carry out atomic spectroscopy of several single- and two-photon transitions of normal hydrogen atoms and comparing the results with QED calculations. The results of the two vastly different experimental approaches appeared to produce robustly consistent results. It therefore came as a great surprise when a measurement of the 2S-2P Lamb shift of muonic hydrogen atoms yielded a proton charge radius that was 4% smaller, corresponding to a discrepancy of between 3 and 5 standard deviations from the earlier results. The highly precise result of this manuscript is interesting and important in the sense that the charge radius determined from the normal and muonic hydrogen atoms now agree. There are a number of high-quality and recent measurements or analysis carried out by other independent groups, however, (e.g., Brandt et al., PRL 128, 023001 (2022) and Fleurbaey et al., PRL 114, 100405 (2018)) that estimate a significantly larger proton charge radius.

These experiments are of the highest importance to bound-state QED which deals with the electromagnetic interaction between two charged particles that lies at the heart of atomic and molecular structure. The experimental uncertainties achieved by combining this work with the authors' previous 1S-2S laser spectroscopy results now reach the same level as the so-called muonic and hadronic vacuum polarization corrections that theoretically arise from, e.g., the effects of virtual hadron-antihadron or muon-antimuon pairs produced in the strong electric field near the atomic nucleus. The high-precision spectroscopy and comparisons with QED calculations may also eventually become sensitive to physics beyond the standard model, such as hypothetical exotic interactions mediated by undiscovered bosons.

The main difficulty of the experiment lies in determining the centroid of the spectroscopic lineshape with an extraordinary precision of one part in 15000 relative to the linewidth. The results of this analysis procedure are significantly affected by even a slight error or oversight in the theoretical modeling of the asymmetric lineshape, and this constitutes the major and enabling advancement of this work. All the known systematic effects such as quantum interference, light force, and dc Stark shifts have been vigorously studied and discussed. This does not exclude the possibility of unknown systematic effects that could affect the results, as the authors are entering into uncharted territory for this type of single-photon laser spectroscopy. Appropriate credit has been given to previous work.

There are several serious issues which should be addressed prior to publication, particularly regarding the title and some of the claims made in the paper:

1: I fully accept the statement of the authors that this work in conjunction with the earlier 1S-2S experimental results constitutes the most precise test of bound-state QED to date for this type of simple atom. In my opinion, however, the additional claim of having tested the standard model to sub-part per trillion precision appearing so prominently in the title, abstract, and conclusions may be a bit misleading particularly in the context of the general readership of Nature. I would suggest that the title, abstract, and some of the conclusions be changed. The term "standard model" appears more than 20 times in the manuscript and figures, but most of these should preferably be replaced with "bound-state QED". The main reason is that the binding energy of the hydrogen atom described by Eq. (1) of the manuscript is expressed in terms of QED (the essential parts of the theory being completed by the early 1950's) which constitutes only a very narrow subset of the standard model that was completed in the 1970's. The standard model is a quantum field theory that describes the interactions between 6 types of quarks, 3 generations of charged leptons and neutrinos, the photon, gluon, and W and Z gauge fields, and the scalar Higgs field. The kinematic properties of these fields may be expressed by a single Lagrangian that contains the electroweak, QCD, Higgs, and Yukawa interaction terms and between 19 and 26 irreducible parameters. At low energies, spontaneous symmetry breaking causes the electroweak force to separate into distinct weak and electromagnetic interactions, so that while QED is often considered the most precise physical theory known to us, it is also but a low-energy effective field theory form of the underlying electroweak theory.

Since the mid-1970's, the term "precision tests of the standard model" when used without any qualification usually refers to measurements of the properties of the neutral and charged weak currents and the W and Z bosons to test the validity of the Glashow-Weinberg (to whom the term "standard model" is sometimes attributed)-Salam model as the correct theory of this

electroweak force, as distinctly separate from the earlier QED. For this reason and in the context of this historical background, the statements in the manuscript such as “The Standard Model prediction of the transition frequency is in excellent agreement with our result, testing the Standard Model to 0.7 parts per trillion” without qualification can confuse the reader, since while the weak interaction mediated by the Z boson affects the hydrogen energies, the contribution is too small (<0.1 kHz ?) to be resolved so far in the present 2S-6P experiment-theory comparison. This leaves us unable to “test” the validity of the electroweak Lagrangian by this experiment alone, though it may become possible in the future with theoretical advances.

It is true that with the discovery of the Higgs boson and the apparent confirmation of most of the remaining parts of the standard model, “precision tests of the standard model” has recently come to include other types of experiments besides the traditional electroweak ones. I believe, however, that it is still problematical to call this experiment “a standard model test with 0.7 ppt precision” for a second important reason: while the electron and muon masses (or almost equivalently, the Yukawa couplings of the electron or muon to the Higgs field with appropriate corrections) are among the irreducible parameters of the standard model, the proton mass m_p that appears in Eq. 1 is not. Owing to the non-perturbative nature of QCD and other issues, m_p cannot be easily and precisely calculated from the standard model Lagrangian though in principle this should be fully possible using the Yukawa couplings of the constituent quarks, the Higgs vacuum expectation value, and the strong, electromagnetic, and weak coupling constants, etc. The most advanced lattice QCD calculations of the proton mass presently have a precision of a few percent or less which corresponds to a theoretical uncertainty on the hydrogen energy of at least 10^{-5} rather than 0.7 ppt if we attempted to calculate Eq. 1 from truly first principles using the standard model Lagrangian alone. As the authors are effectively using the experimental values of the proton-to-electron mass ratio, proton charge radius, magnetic moment, etc. as either the input parameters of the bound-state QED calculation or in the derivation of the 2S-6P transition frequency, instead of calculating them from the standard model including the QCD and electroweak parts, the test is largely restricted to QED. This restriction is a benefit in some sense (see below).

2: In the important result of Eq. 10, the left hand side apparently designates a standard-model prediction $\nu_{2S-6P, SM}$ whereas the right-hand side designates a QED calculation and the uncertainty on the measured (not calculated by QCD) proton charge radius, $730\,690\,248\,610.79(18)_{QED(14)_{rp}}$ kHz. I believe “QED” should not be used interchangeably with “standard model” for the historical-conventional and important theoretical reasons outlined above. Indeed, in several places in the manuscript the authors correctly state that “Until now, this has prevented a verification of QED at the level of experimental uncertainties.”

Suppose we in the distant future acquire the theoretical tools that allows a precise calculation of the proton mass using the interactions described by the standard model, and it was found that a new previously unappreciated feature of the strong or electroweak interaction was causing the experimental value of m_p to deviate from the standard model prediction at a level of 10^{-9} . As far as I know this possibility is not yet excluded. The “test” of Eq. 10 would still be correct if it were restricted to bound state QED in the context of atomic physics because the theory allows us to calculate the hydrogen energies to the ppt scale regardless; but things may arguably be more difficult to reconcile if the same Eq. 10 were interpreted as a standard model test with a calculated m_p value. This distinction is important as it leaves the door open for future discoveries if our ability to solve the standard model Lagrangian were to become sufficiently advanced.

3: The quantum numbers F and m_F appear in the main text without being defined.

4: The notations for the transitions such as $2S-6P_{1/2}$ and $2S-6P_{3/2}$ (with the quantum number $J=1/2$ indicated for the final state but not the initial state) may be difficult for non-specialists to follow: the “2S” in the “2S-6P” transition corresponds to the hyperfine structure (HFS) centroid of the 2S manifold, while for the “ $2S-6P_{1/2}$ ” transition the notation “2S” used by the authors apparently refers to the magnetic substate $2S^{F=0}_{1/2}$, $m_F=0$ instead of the HFS centroid as in the “2S-6P” case. To avoid this possible inconsistency, perhaps the notations could be changed or better explained at the beginning of the paper along with the definitions of the quantum numbers, instead of in the Methods section. A brief sentence in the main text explaining the selection rules in the context of why these two particular transitions were chosen in the experiment may be useful for the non-specialist, though the details are available in the Methods section.

5: Along the same vein, the terms “HFS centroid” and “fine structure centroid” should be briefly explained in the main text since they directly relate to the 2S-6P transition frequency being measured, though it is mentioned in the Methods section and defined in Eq. 7. Why is the fine-structure centroid particularly interesting instead of some other transition frequency between the 2S and 6P manifolds?

6: Similarly, it may be difficult to understand why the title of Section 5.1 is in plural form “2S-6P transition frequencies”, whereas in the text only a single frequency for 2S-6P is determined.

7: In Section 1, the authors state “This precision is enabled by a detailed understanding of the line shape and a large signal-

to-noise ratio, allowing the determination of the transition frequency to one part in 15000 of the experimental linewidth."

-> A few sentences regarding the significance and substantial difficulty of absolutely determining the centroid of a spectral lineshape to one part in 15000 relative to the linewidth is needed here. The non-specialist reader should be made aware that the results of this type of analysis procedure would be significantly affected by even a slight error or omission in the theoretical modelling of the lineshape, and that this constitutes the major and enabling advancement of this work. All known systematic effects that could conceivably cause a spurious shift in the experimental results have been vigorously studied as far as I could see; nevertheless the authors are still treading into unexplored territory, so there is significant risk of an oversight that could affect the results or increase the uncertainties.

8: In Fig. 1, it is difficult to tell which data point was taken from which cited work, though the unique publication year provides a clue. The reader may want to distinguish the measurements carried out by independent groups; this is particularly important in view of the large deviations between the experimental results.

9: The authors state, "Subsequent measurements in atomic hydrogen have followed [3–6, 14], but they are partly discrepant with the muonic value and with each other, and none is precise enough to conclusively confirm the muonic value, as visualized in Fig. 1. Until now, this has prevented a verification of QED at the level of experimental uncertainties."

-> Perhaps this can be slightly reworded, as it is not the a priori purpose of a high-precision experiment to conclusively confirm an existing result. If multiple groups continue to carry out a series of experiments that each have statistical and systematic uncertainties and fluctuations that are difficult to reliably quantify (thus presumably giving rise to the statistically improbable 3 to 5 sigma deviations in the experimental values of the proton charge radius) until one fluctuating outcome happens to coincide with another one, and at that point we claim "conclusive confirmation", this is a procedure almost guaranteed to converge but not necessarily to the true physical value. It is vitally important to note that at least two other independent experimental groups (e.g., Brandt et al., PRL 128, 023001 (2022) and Fleurbaey et al., PRL 114, 100405 (2018)) continue to estimate significantly larger proton charge radii that are in tension with the muonic hydrogen value.

10: In Section 2.1 the authors state "We study the 2S-6P transition in a cryogenic beam of hydrogen atoms using Doppler-free one-photon laser spectroscopy."

-> Is it really appropriate to call the 410 nm single-photon transition "Doppler-free"? The resonance lineshape of Fig. 2 is clearly Doppler-broadened by the transverse velocity distribution of the hydrogen beam, and it is only when the atoms are very carefully irradiated by two counter-propagating laser beams of equal intensity and similar spatial wavefronts that the centre-of-gravity of the resulting Doppler-broadened resonance line effectively does not shift. This is in fundamental contrast to the 1S-2S two photon resonance lineshape where the Doppler broadening effect does not appear to first order and so the method is intrinsically "Doppler free".

11: In the caption to Fig. 2 (c) we read, "The FWHM (full width at half maximum) linewidth Γ_F reduces for slower velocity groups as Doppler broadening reduces".

-> The experimental data shows Doppler broadening when the measurement was said to be "Doppler-free" in Section 1 (see above), perhaps the term "Doppler-free" should not be used as it may confuse the reader.

12: In the spectra of Fig. 2(c), the error bars in the plot are too small, making it difficult to see how consistent they are with the simulated lineshape; the goodness-of-fit is vitally important in an experiment that determines the centre-of-gravity of the lineshape to 1 part in 15000 of the linewidth. It would therefore be extremely useful to plot the error residuals ([Exp. data]-[Simulation]) including the error bars in a separate plot beneath Fig. 2(c).

13: In Fig. 2(c), the wings of each of the three spectral profiles seem to be defined by only two data points at detuning frequencies of -22 MHz and +22 MHz. A symmetrically equal number of datapoints are positioned at the low-frequency (left) and high-frequency (right) sides of each peak at detunings <10 MHz very close to the peak. In other experiments, such a highly non-equidistant choice of data points might be generally considered a risky data-taking strategy: if we choose the positions of the data points to most efficiently resolve the characteristics of a pre-existing model (in this case the centroid), we are in fact increasing the risk of biasing the outcome to agree with our expectations since this dataset choice arguably reduces the chances of detecting a significant deviation between the experiment and simulated lineshape. From this spectra alone we cannot tell how well the theory reproduces the spectral shape at larger laser detunings; perhaps there might be a small constant offset or other unexpected structure superimposed on the Voigt-like function?

14: "The excited 6P levels rapidly decay, directly or through cascades, to the 1S and 2S manifolds, resulting in a $\Gamma = 3.89$ MHz natural transition linewidth."

-> I summed the single-photon transition rates of the relativistic calculations tabulated for example in W.L. Wiese and J.R. Fuhr J. Phys. Chem. Ref. Data 38, 565-726 (2009), and got a number like 3.90 MHz instead of 3.89 MHz though I am not confident about this number.

15: The authors state " $\gamma_{el}/\Gamma = 3.9\%$ or 7.9% of decays from the excited level lead back to the initial 2S level for the 2S-6P_{1/2} or 2S-6P_{3/2} transitions, respectively."

-> Perhaps the sentence should not be started by an equation. I suppose by "initial 2S level" you actually mean the 2S^{F=0}_{1/2}, m_F=0 sublevel?

16: "The 6P fine-structure splitting $\Delta\nu_{FS}(6P) \approx 405$ MHz between the two excited levels"

-> Four levels are indicated for the 6P manifold in Fig. 2 which may be confusing for some in the general readership. Perhaps the uncertainty of the 405 MHz value can be indicated in this sentence.

17: "Nevertheless, quantum interference (QI) between excitation–decay paths through either excited fine-structure manifold can lead to substantial distortions of the line shape"

-> The non-specialist may be baffled by the adverb "nevertheless". A very brief explanation of the quantum interference effect, its dependency on the natural width and spectral spacings of multiple resonance lines, and why it may potentially affect the resonance lineshape is required here. Also needed are the reasons why the quantum interference is suppressed at the magic angle of 56.5 degrees of the laser polarization angle and by the increased solid angle of the detector, as these details are mentioned later in the main text without an accessible explanation. A read of Section 4 also provides no explanation of what quantum interference might be.

18: In Section 2.2, the authors state "...offset angle $|\alpha_0| = 0$ mrad . . . 12 mrad from the orthogonal". Later we read that "We align α_0 close to zero".

-> At first read it is hard to interpret whether the indicated range of angles is caused by the divergence of the atomic and laser beams (mentioned in the previous paragraph of the main text), or represent the offset angles that were intentionally scanned to evaluate systematic uncertainties. Later in the paper and in Methods we find that the latter is true.

19: "wavefront-retracing"

-> This may be jargon; I could only find a few groups that use this terminology in publications searchable by Google Scholar. It should be briefly explained since it constitutes a vital point in the experimental work. What happens when the wavefront is not "retraced"?

20: The term "dc Stark shifts" is mentioned several times in the main text without explaining what they are in the main text. A brief sentence explaining why the shift has (mostly) a quadratic dependence to the electric field may aid the reader.

21: In Section 2.3 we read "distortions from the LFS dominate, for which no equivalent line shape model is known to us"

-> The acronym LFS appears without explanation, though it is defined later in Section 3.

22: Perhaps some of the highly interesting but detailed explanations of Section 3 can be moved to either Methods or Supplementary Materials, to make space for other vital explanations that would make the paper more accessible to the general reader.

23: "This precision is comparable to the test of the Standard Model prediction of the electron magnetic moment, which is currently limited to 0.7 ppt by discrepant measurements of α [11]."

-> This may be a matter of taste, but a measurement of the g-2 anomalous magnetic moment of a single lepton, and a measurement of a two-body lepton-hadron bound system, are probing related but quite different things and so a comparison of the relative precision may not be so informative. Of greater importance is the fact that both the electron g-2 and hydrogen experiments and theory reach the unprecedented precision of three-photon corrections.

24: In Methods we read, "which we attribute to fluctuations in the atomic flux (of metastable 2S atoms)"

-> What is the reason for the parenthesis?

25: "more refined spatial detection efficiency simulations"

-> 5 modifiers/adjectives

26: "Importantly, while Eq. (21) is found to approximately hold in either regime, the coefficient β_{dc} may differ."

-> A brief sentence explaining why the coefficient is allowed to differ in the two experimental regimes may be useful. What happens if we insist that the coefficient be the same in both regimes, as may be naively expected?

Referee #3

(Remarks to the Author)

The manuscript under review addresses the measurement of the 2S–6P transition in atomic hydrogen. Spectroscopic measurements in light atoms provide the most precise low-energy tests of the Standard Model of fundamental interactions, with the hydrogen atom being the most prominent case. Together with the extremely accurate 1S–2S transition measurement, one additional high-precision transition is required in order to extract both the Rydberg constant and the proton charge radius, on which binding energies depend. The authors accomplish exactly this: their measurement of the 2S–6P transition is so precise that the resulting value of the proton charge radius surpasses the accuracy of all previous determinations from atomic hydrogen. Moreover, it is in excellent agreement with the value obtained from muonic hydrogen. This is a very significant result, given the longstanding discrepancies in proton charge radius determinations from atomic hydrogen. The results presented in this work are therefore of broad importance to the physics community and, in my opinion, merit publication in Nature.

Authors are experienced and respected experimentalists with a long record of measurements on hydrogenic systems (both ordinary and muonic). The present work represents a continuation of their previous efforts in the field (in this case their 2S–4P measurement, which is cited in the manuscript). The manuscript consists of the main section, a methodology section, and the Supplemental Material. The main section provides a clear description of the principle of experiment. The most important contributions affecting the determination of resonance frequencies (Doppler effect, quantum interference and light force shift corrections) are briefly discussed, with detailed treatments deferred to the Methods section and the Supplemental Material. The results obtained for the two transition frequencies into 6P fine structure levels are highly impressive and constitute exceptionally accurate test of the Standard Model predictions. The conclusions drawn in the manuscript are valid and well supported.

The methodology section presents specific details of the experiment, including statistics, modeling and data analysis. The presentation is thorough, transparent and the discussion of error bars is sound and well justified. All data (including the tables and figures in Appendix) are presented clearly. The Supplemental Material provides further details of the models used in the measurement and the interpretation of results, which I find satisfactory. It also includes a discussion of less significant corrections that are included in table A3. Overall, the manuscript is clearly written and accessible.

To summarize my report: I recommend the publication of this manuscript in Nature. The transition frequencies reported in this work allow for extremely accurate test of the Standard Model (on level of ppt) and yield the most accurate determination of the proton charge radius from atomic hydrogen to date. This result is in excellent agreement with the corresponding value from muonic hydrogen and contributes to resolving the proton radius puzzle. For these reasons, I believe this work fully meets the criteria for publication in this journal.

Minor correction: in Supplemental Material, on line 233: "which" should probably be "with"

Response to referee #1

Referee comments are italicized; our (the authors') responses are upright and in blue.

The manuscript entitled "Sub-part-per-trillion test of the Standard Model with atomic hydrogen" presents results from a thorough and high-level experimental work.

This work concerns the measurement of the 2S-6P transition frequency in hydrogen atoms with a relative uncertainty of 6.6×10^{-13} , leading to the determination of the proton radius with a relative uncertainty of 1.8×10^{-3} , a bit more than 2.5-fold compared to other determinations from hydrogen spectroscopy. This result confirms definitively the value obtained by muon spectroscopy and closes, in my opinion, definitively the debate on the famous proton puzzle. The remarkable agreement with the Standard Model (SM) prediction using the value of the proton radius (from the muon) constitutes a precise test of the SM and QED calculations, almost competitive with the test using the electron's $g-2$.

The experiment is described in great details. The exhaustive analysis of several noise sources and systematic biases presented in "The Method" and "The Supplementary Method" reflects both the in-depth and rigor of the experimental methodology. I have reread the manuscript several times and it is difficult to find any major criticisms, as this paper is written with remarkable clarity and precision.

I have only two suggestions to help clarify some systematic effects

1) The quantum interference effect and the concept of the magic polarization angle are briefly mentioned in the main text and refer only to two references that may be difficult for non-specialists to understand. Even though the evaluation of the related systematic effect is detailed in the Method and Supplementary Method, the document would gain in clarity if the authors took the time to provide an intuitive explanation of this magic polarization angle and its link to quantum interference (in few sentences).

We've expanded the brief discussion of QI at the end of Section 2.1, which now explains the underlying physical process, the connection of the order of magnitude of the associated shifts with the linewidth and fine-structure splitting, and the dependence of the shift on the detection geometry and laser polarization angle. We've also made some small edits to the relevant sentence in Section 1.2 of the Supplementary Methods.

2) In S2.8 Recoil shift, authors may add a sentence indicating why Gouy phase and wavefront curvature are neglected.

We have added a sentence in S2.8 stating that corrections to the recoil shift from wavefront curvature and Gouy phase amount to at most 0.1 Hz and can therefore be neglected.

In conclusion, I strongly recommend the publication of this paper and congratulate the authors on their excellent work.

Response to referee #2

Referee comments are italicized; our (the authors') responses are upright and in blue.

In the manuscript "Sub-part-per-trillion test of the Standard Model with atomic hydrogen" by Maisenbacher et al., the authors determine two transition frequencies of hydrogen atoms contained in a cryogenic beam to a precision of 0.9 parts per trillion using single-photon laser spectroscopy. The first transition was excited between a hyperfine magnetic substate of the metastable 2S manifold with a total electronic angular momentum quantum number of $J=1/2$, total angular momentum quantum number $F=0$, and total angular momentum magnetic quantum number $mF=0$ (denoted as $2SF=01/2$, $mF=0$); and a substate of the 6P manifold with the quantum numbers $J=1/2$, $F=1$, and $mF=0$ ($6SF=11/2$, $mF=0$). The second transition was excited between the substates $2SF=01/2$, $mF=0$ and $6SF=13/2$, $mF=0$. From this the authors calculated the so-called 2S-6P transition frequency between the hyperfine centroid of the 2S manifold and the fine-structure centroid of the 6P manifold as 730690248610.79(48) kHz. By comparing the results with quantum electrodynamics calculations and the 1s-2s transition frequency measured by the same group in a previous publication, the root-mean-square charge radius of the proton was determined as 0.8406(15) fm. This result is 2.5 times more precise than any previous determination based on spectroscopy of hydrogen while also coinciding with the charge radius determined by laser spectroscopy of the 2S-2P transition of muonic hydrogen atoms. The value significantly disagrees with the CODATA 2014 value of the proton charge radius.

I recommend publication of this work in Nature due to its central importance in testing the validity of quantum electrodynamics (QED) which forms the basis of the standard model of particle physics and specifically in resolving the so-called "proton radius puzzle". The spatial distribution of electric charge within a proton is affected by the complex dynamics of the valence and virtual sea quarks and of the gluons that mediate the strong interaction. The proton charge radius cannot yet be easily calculated from first principles using quantum chromodynamics (QCD) because of the non-perturbative nature of the strong interaction at the low momentum transfer regions relevant to the inner structure of the proton. So far, the only ab initio way of calculating the electric and magnetic radii has been provided by lattice QCD methods containing numerous simplifying approximations. Prior to 2010, the experimental values of the charge radius were determined either by scattering high energy beams of electrons on proton targets and carrying out a phenomenological analysis to extrapolate the scattering data to zero momentum transfer, or by carry out atomic spectroscopy of several single- and two-photon transitions of normal hydrogen atoms and comparing the results with QED calculations. The results of the two vastly different experimental approaches appeared to produce robustly consistent results. It therefore came as a great surprise when a measurement of the 2S-2P Lamb shift of muonic hydrogen atoms yielded a proton charge radius that was 4% smaller, corresponding to a discrepancy of between 3 and 5 standard deviations from the earlier results. The highly precise result of this manuscript is interesting and important in the sense that the charge radius determined from the normal and muonic hydrogen atoms now agree. There are a number of high-quality and recent measurements or analysis carried out by other independent groups, however, (e.g., Brandt et al., PRL 128, 023001 (2022) and Fleurbaey et al., PRL 114, 100405 (2018)) that estimate a significantly larger proton charge radius.

These experiments are of the highest importance to bound-state QED which deals with the electromagnetic interaction between two charged particles that lies at the heart of atomic and molecular structure. The experimental uncertainties achieved by combining this work with the authors' previous 1S-2S laser spectroscopy results now reach the same level as the so-called

muonic and hadronic vacuum polarization corrections that theoretically arise from, e.g., the effects of virtual hadron-antihadron or muon-antimuon pairs produced in the strong electric field near the atomic nucleus. The high-precision spectroscopy and comparisons with QED calculations may also eventually become sensitive to physics beyond the standard model, such as hypothetical exotic interactions mediated by undiscovered bosons.

The main difficulty of the experiment lies in determining the centroid of the spectroscopic lineshape with an extraordinary precision of one part in 15000 relative to the linewidth. The results of this analysis procedure are significantly affected by even a slight error or oversight in the theoretical modeling of the asymmetric lineshape, and this constitutes the major and enabling advancement of this work. All the known systematic effects such as quantum interference, light force, and dc Stark shifts have been vigorously studied and discussed. This does not exclude the possibility of unknown systematic effects that could affect the results, as the authors are entering into uncharted territory for this type of single-photon laser spectroscopy. Appropriate credit has been given to previous work.

There are several serious issues which should be addressed prior to publication, particularly regarding the title and some of the claims made in the paper:

1: I fully accept the statement of the authors that this work in conjunction with the earlier 1S-2S experimental results constitutes the most precise test of bound-state QED to date for this type of simple atom. In my opinion, however, the additional claim of having tested the standard model to sub-part per trillion precision appearing so prominently in the title, abstract, and conclusions may be a bit misleading particularly in the context of the general readership of Nature. I would suggest that the title, abstract, and some of the conclusions be changed. The term "standard model" appears more than 20 times in the manuscript and figures, but most of these should preferably be replaced with "bound-state QED". The main reason is that the binding energy of the hydrogen atom described by Eq. (1) of the manuscript is expressed in terms of QED (the essential parts of the theory being completed by the early 1950's) which constitutes only a very narrow subset of the standard model that was completed in the 1970's. The standard model is a quantum field theory that describes the interactions between 6 types of quarks, 3 generations of charged leptons and neutrinos, the photon, gluon, and W and Z gauge fields, and the scalar Higgs field. The kinematic properties of these fields may be expressed by a single Lagrangian that contains the electroweak, QCD, Higgs, and Yukawa interaction terms and between 19 and 26 irreducible parameters. At low energies, spontaneous symmetry breaking causes the electroweak force to separate into distinct weak and electromagnetic interactions, so that while QED is often considered the most precise physical theory known to us, it is also but a low-energy effective field theory form of the underlying electroweak theory.

Since the mid-1970's, the term "precision tests of the standard model" when used without any qualification usually refers to measurements of the properties of the neutral and charged weak currents and the W and Z bosons to test the validity of the Glashow-Weinberg (to whom the term "standard model" is sometimes attributed)-Salam model as the correct theory of this electroweak force, as distinctly separate from the earlier QED. For this reason and in the context of this historical background, the statements in the manuscript such as "The Standard Model prediction of the transition frequency is in excellent agreement with our result, testing the Standard Model to 0.7 parts per trillion" without qualification can confuse the reader, since while the weak interaction mediated by the Z boson affects the hydrogen energies, the contribution is too small (<0.1 kHz ?) to be resolved so far in the present 2S-6P experiment-

theory comparison. This leaves us unable to "test" the validity of the electroweak Lagrangian by this experiment alone, though it may become possible in the future with theoretical advances.

It is true that with the discovery of the Higgs boson and the apparent confirmation of most of the remaining parts of the standard model, "precision tests of the standard model" has recently come to include other types of experiments besides the traditional electroweak ones. I believe, however, that it is still problematical to call this experiment "a standard model test with 0.7 ppt precision" for a second important reason: while the electron and muon masses (or almost equivalently, the Yukawa couplings of the electron or muon to the Higgs field with appropriate corrections) are among the irreducible parameters of the standard model, the proton mass m_p that appears in Eq. 1 is not. Owing to the non-perturbative nature of QCD and other issues, m_p cannot be easily and precisely calculated from the standard model Lagrangian though in principle this should be fully possible using the Yukawa couplings of the constituent quarks, the Higgs vacuum expectation value, and the strong, electromagnetic, and weak coupling constants, etc. The most advanced lattice QCD calculations of the proton mass presently have a precision of a few percent or less which corresponds to a theoretical uncertainty on the hydrogen energy of at least 10^{-5} rather than 0.7 ppt if we attempted to calculate Eq. 1 from truly first principles using the standard model Lagrangian alone. As the authors are effectively using the experimental values of the proton-to-electron mass ratio, proton charge radius, magnetic moment, etc. as either the input parameters of the bound-state QED calculation or in the derivation of the 2S-6P transition frequency, instead of calculating them from the standard model including the QCD and electroweak parts, the test is largely restricted to QED. This restriction is a benefit in some sense (see below).

We thank the referee for this detailed comment and follow their argument that the term Standard Model tests previously described tests of the electroweak sector in the particle physics community. However, as the referee points out, "“precision tests of the standard model” has recently come to include other types of experiments besides the traditional electroweak ones". While it is true that we test a certain sector of the Standard Model – the low-energy sector of electromagnetic interaction, which is described by QED – the same is true for electroweak theory tests, and any other test of the Standard Model (such as precision symmetry tests or precision neutrino oscillation tests). We would like to emphasize that our measurement can also be used to improve constraints on beyond-Standard-Model weakly interacting bosons with masses in the keV range (see Delaunay et al., Phys. Rev. Lett. 130, 121801 (2023)), as noted in the second-to-last paragraph in the main text.

We therefore think it appropriate to call our results a test of the Standard Model (we respectfully note that we view QED as a foundational rather than narrow subset of the Standard Model). This language has been prominently applied to other precision measurements in the low-energy electromagnetic regime, such as the measurement of the muon magnetic moment (Phys. Rev. Lett. 135, 101802 (2025); first sentence "Precise measurements of magnetic moments of charged leptons serve as precision probes of the Standard Model (SM)."), the electron magnetic moment (Fan et al., Phys. Rev. Lett. 130, 071801 (2023); abstract: "The most precisely determined property of an elementary particle tests the most precise prediction of the standard model (SM) to 1 part in 10^{12} ."), and the fine-structure constant (Parker et al., Science 360, 191 (2018); title: "Measurement of the Fine Structure Constant as a Test of the Standard Model"). Furthermore, we note that the other referees specifically highlight the importance of our results as a test of the Standard Model.

Therefore, we do not think that calling our experiment a test of the Standard Model may confuse the reader but instead helps the reader to readily identify our experiment as a member of a class of well-known, widely discussed measurements (with the addition of "[...] with atomic hydrogen" giving experts in the field additional context). In addition, we identify the subset of the

Standard Model we are testing in our abstract (which we have slightly edited to further highlight the relationship between QED and the Standard Model).

Regarding the referee's point about the proton mass not being calculable from first principles within the Standard Model to sufficient precision, we address this in our response to the referee's second comment below.

2: In the important result of Eq. 10, the left hand side apparently designates a standard-model prediction $v_{2S-6P, SM}$ whereas the right-hand side designates a QED calculation and the uncertainty on the measured (not calculated by QCD) proton charge radius, $730\,690\,248\,610.79(18)_{QED(14)}r_p$ kHz. I believe "QED" should not be used interchangeably with "standard model" for the historical-conventional and important theoretical reasons outlined above. Indeed, in several places in the manuscript the authors correctly state that "Until now, this has prevented a verification of QED at the level of experimental uncertainties."

Suppose we in the distant future acquire the theoretical tools that allows a precise calculation of the proton mass using the interactions described by the standard model, and it was found that a new previously unappreciated feature of the strong or electroweak interaction was causing the experimental value of m_p to deviate from the standard model prediction at a level of 10^{-9} . As far as I know this possibility is not yet excluded. The "test" of Eq. 10 would still be correct if it were restricted to bound state QED in the context of atomic physics because the theory allows us to calculate the hydrogen energies to the ppt scale regardless; but things may arguably be more difficult to reconcile if the same Eq. 10 were interpreted as a standard model test with a calculated m_p value. This distinction is important as it leaves the door open for future discoveries if our ability to solve the standard model Lagrangian were to become sufficiently advanced.

As we have discussed above, our position is that a test of the low-energy electromagnetic sector of the Standard Model, described by QED, constitutes a test of the Standard Model. Conversely, not every test of the Standard Model is a test of QED.

However, we do not use the terms interchangeably: we refer to the overall Eq. (1) as the Standard Model prediction but then break out different sources of known uncertainty. The contribution labeled "QED" is from the known uncertainty of QED calculations, while the uncertainty labelled " r_p " is from the known uncertainty of the proton radius.

The point made by the referee is a good example for why we call it a test of the Standard Model: if a new, precise prediction for m_p was made, and the theory prediction of Eq. (1) using this value of m_p were to disagree with our experimental result, we would conclude that either this new m_p value is incorrect or there are, coincidentally, other missing terms that have so far masked this discrepancy (others might conclude our measurement is wrong). We would then have tested the strong or electroweak sector of the Standard Model through its effect on the hydrogen spectrum. Our aim is to make things difficult to reconcile if new features are found, thereby allowing for the discovery of such features.

As another example, one may insert a calculated value of the proton radius r_p in Eq. (1) and compare with our result. r_p can be calculated from first principles using quantum chromodynamics (QCD), a sector of the Standard Model distinct from QED, with calculations currently only about one order of magnitude less precise than required (see reference 19 of our manuscript). This test of QCD may therefore be possible in the near future.

3: The quantum numbers F and mF appear in the main text without being defined.

Thank you, we have added the definitions " F : total angular momentum quantum number, mF : magnetic quantum number" in Section 2.1 and renamed the quantum number J from "total

angular momentum quantum number" to "total electronic angular momentum quantum number" (see Section 1).

4: The notations for the transitions such as 2S-6P1/2 and 2S-6P3/2 (with the quantum number $J=1/2$ indicated for the final state but not the initial state) may be difficult for non-specialists to follow: the "2S" in the "2S-6P" transition corresponds to the hyperfine structure (HFS) centroid of the 2S manifold, while for the "2S-6P1/2" transition the notation "2S" used by the authors apparently refers to the magnetic substate $2S_{F=0, mF=0}$ instead of the HFS centroid as in the "2S-6P" case. To avoid this possible inconsistency, perhaps the notations could be changed or better explained at the beginning of the paper along with the definitions of the quantum numbers, instead of in the Methods section. A brief sentence in the main text explaining the selection rules in the context of why these two particular transitions were chosen in the experiment may be useful for the non-specialist, though the details are available in the Methods section.

We have reworded the sentence defining the transitions and our notation, and hope this now reads more clearly.

We kept the notation of "2S-6P1/2" and "2S-6P3/2" to label the two measured transitions, as we believe adding more quantum numbers to the labels will negatively affect the readability of the text, especially for non-experts. Likewise, we'd rather leave the discussion of selection rules to the Methods section as to not overload the text with not immediately relevant details.

We note that the term "2S-6P transition" is used when referring to shared properties of the two transitions and does not refer to specific levels.

5: Along the same vein, the terms "HFS centroid" and "fine structure centroid" should be briefly explained in the main text since they directly relate to the 2S-6P transition frequency being measured, though it is mentioned in the Methods section and defined in Eq. 7. Why is the fine-structure centroid particularly interesting instead of some other transition frequency between the 2S and 6P manifolds?

The 2S-6P fine-structure centroid is briefly defined when it is first mentioned in Section 4 ("Quantum interference"), which precedes Eq. (7). This section also explains why it is particularly interesting, as it strongly suppresses the quantum interference effect: "In addition, we make use of the opposite signs and relative strengths of the shifts and combine the 2S-6P1/2 and 2S-6P3/2 transition frequencies with a 1:2 ratio into the 2S-6P fine-structure centroid (see Methods), which reduces the shift to an insignificant $-0.05(2)$ kHz."

Currently, the term "HFS centroid" does not appear in the main text (but is used in the explicit definition of the 2S-6P fine-structure centroid in the Methods and one might argue therefore implicitly appears in the main text).

We believe a more explicit definition of the 2S-6P fine-structure centroid, including the HFS centroid, in the main text (as given in the Methods) will degrade the readability of the text while adding little benefit, especially for non-expert readers. We therefore have not made any changes to address this comment.

6: Similarly, it may be difficult to understand why the title of Section 5.1 is in plural form "2S-6P transition frequencies", whereas in the text only a single frequency for 2S-6P is determined.

Section 5.1 presents two transition frequencies, $\nu_{1/2}$ and $\nu_{3/2}$, of the two fine-structure-resolved transitions (2S-6P1/2 and 2S-6P3/2) that were measured in the experiment. These two transition frequencies are then later combined into the 2S-6P fine-structure centroid to reduce

certain systematic effects (see above). Hence, we believe "2S-6P transition frequencies" is appropriate. We therefore have not made any changes to address this comment.

7: In Section 1, the authors state "This precision is enabled by a detailed understanding of the line shape and a large signal-to-noise ratio, allowing the determination of the transition frequency to one part in 15000 of the experimental linewidth."

-> A few sentences regarding the significance and substantial difficulty of absolutely determining the centroid of a spectral lineshape to one part in 15000 relative to the linewidth is needed here. The non-specialist reader should be made aware that the results of this type of analysis procedure would be significantly affected by even a slight error or omission in the theoretical modelling of the lineshape, and that this constitutes the major and enabling advancement of this work. All known systematic effects that could conceivably cause a spurious shift in the experimental results have been vigorously studied as far as I could see; nevertheless the authors are still treading into unexplored territory, so there is significant risk of an oversight that could affect the results or increase the uncertainties.

We modified the sentence in question to read: "This precision corresponds to finding the transition frequency to one part in 15,000 of the experimental linewidth, to our knowledge unprecedented for laser spectroscopy, requiring a thorough understanding of any asymmetric distortions of the line shape at that level and a large experimental signal-to-noise ratio." The sentence now highlights the need to understand asymmetric line shape distortions at the level of the achieved line splitting and directly states that this level of line splitting is unprecedented in absolute frequency determinations using laser spectroscopy (in microwave spectroscopy, higher line splittings are routinely achieved: the 2019 hydrogen 2S-2P measurement (Bezginov et al., Science 365, 1007–1012 (2019)) reaches 1 part in 30,000; Cs fountain clocks, upon which the SI second is based, reach 1 part in a million).

However, we would like to note that any experiment that is limited in uncertainty by corrections relating to the line shape will be significantly affected by even a slight error in line shape modeling. This is true independent of the line splitting. One may expand the line shape into an unperturbed (symmetric) part (Lorentzian or Voigt) and a perturbed part, where one is mostly interested in asymmetric perturbations. Then, one should understand the perturbation on a level corresponding to which level the shift from the perturbation needs to be understood to reach the quoted uncertainty. An experiment with a moderate line splitting of $1e-3$ that however needs to account for line shape distortions to a level of 1 % (e.g., some of the two-photon spectroscopy results) needs a better understanding of their line shape distortion than our experiment, which has a line splitting of $7e-5$, but only needs to account for line shape distortions at the level of 10 %. In fact, the level of frequency corrections related to line shape distortions is comparatively small in our experiment - completely disregarding all these corrections (summed up as "Simulation corrections") only shifts our result by about 2 sigma. In fact, this low level of systematic corrections was one of the main reasons we chose to study one-photon transitions in the first place.

There are, however, some technical challenges with increasing line splitting: e.g., a potential sloped signal offset has a larger impact, and it is numerically more challenging to resolve small line distortions in simulations. We discuss these and other related issues in detail in the Methods and Supplementary Methods, where they are easily accessible to the expert reader without distracting the non-specialist reader.

In addition, one might argue that the systematic effects we encounter in one-photon spectroscopy are less well studied than those in two-photon spectroscopy, that is, that we're "treading into unexplored territory" as the referee puts it. It is for this reason that we devoted a significant portion of both our experimental effort as well as this manuscript to a detailed

experimental study of the light force shift. Likewise, we studied quantum interference in detail in our previous 2S-4P measurement, and use the lessons learned there to strongly suppress it here. For both systematic effects, we implemented powerful experimental cross-checks, and we use the 6P fine-structure splitting as a cross-check covering multiple systematic effects. The analysis was performed while the result was blinded to prevent bias.

All in all, while we cannot exclude the possibility of unaccounted systematic effects, we believe our manuscript in its entirety demonstrates our level of care and the difficulty of the experiment to the reader, and we hope the modified introductory sentence primes the reader for the detailed studies that follow.

8: In Fig. 1, it is difficult to tell which data point was taken from which cited work, though the unique publication year provides a clue. The reader may want to distinguish the measurements carried out by independent groups; this is particularly important in view of the large deviations between the experimental results.

We have added the references corresponding to the data points in Fig. 1 (and Extended Data Fig. 2, showing the Rydberg constant). We will also provide the source data for all figures, including this one (available with the online versions of the publication).

We believe that additionally labeling and discussing the laboratories from which the different measurements stems will overload the figure and text, while inviting a discussion of the origins of the proton radius puzzle that is beyond the scope of this paper. We note that the fact that the 2S-4P result is from our group is clearly stated in the text, while the 2020 1S-3S result, also from our group, stems from a conceptually (and physically) different experiment.

9: The authors state, "Subsequent measurements in atomic hydrogen have followed [3–6, 14], but they are partly discrepant with the muonic value and with each other, and none is precise enough to conclusively confirm the muonic value, as visualized in Fig. 1. Until now, this has prevented a verification of QED at the level of experimental uncertainties."

 Perhaps this can be slightly reworded, as it is not the a priori purpose of a high-precision experiment to conclusively confirm an existing result. If multiple groups continue to carry out a series of experiments that each have statistical and systematic uncertainties and fluctuations that are difficult to reliably quantify (thus presumably giving rise to the statistically improbable 3 to 5 sigma deviations in the experimental values of the proton charge radius) until one fluctuating outcome happens to coincide with another one, and at that point we claim "conclusive confirmation", this is a procedure almost guaranteed to converge but not necessarily to the true physical value. It is vitally important to note that at least two other independent experimental groups (e.g., Brandt et al., PRL 128, 023001 (2022) and Fleurbaey et al., PRL 114, 100405 (2018)) continue to estimate significantly larger proton charge radii that are in tension with the muonic hydrogen value.

We now write "[...] none [of the previous measurements] is precise enough to conclusively test the muonic value [...]" (i.e., "test" instead of "confirm"), where "conclusively" was here defined as >5 sigma in the preceding sentence.

We agree that is important to note that some of the previous measurements are in tension with the muonic value, which is why we prominently include those results in Fig. 1.

We would also like to emphasize that the situation here is different to the one described in the referee's example in one key point, which is that our measurement is much more precise than the previous measurements. An improbable random fluctuation of 5 sigma in our result would still leave us 4.3 sigma below the "large" proton radius (CODATA 2014), while a 5 sigma

fluctuation of any of the other results covers both the muonic and the CODATA 2014 value within at most 1 sigma.

10: In Section 2.1 the authors state “We study the 2S-6P transition in a cryogenic beam of hydrogen atoms using Doppler-free one-photon laser spectroscopy.”

-> Is it really appropriate to call the 410 nm single-photon transition “Doppler-free”? The resonance lineshape of Fig. 2 is clearly Doppler-broadened by the transverse velocity distribution of the hydrogen beam, and it is only when the atoms are very carefully irradiated by two counter-propagating laser beams of equal intensity and similar spatial wavefronts that the centre-of-gravity of the resulting Doppler-broadened resonance line effectively does not shift. This is in fundamental contrast to the 1S-2S two photon resonance lineshape where the Doppler broadening effect does not appear to first order and so the method is intrinsically “Doppler free”.

First, we would like to clarify that we do not call the transition itself “Doppler-free”, but our spectroscopy technique. We write: “We study the 2S-6P transition in a cryogenic beam of hydrogen atoms using Doppler-free one-photon laser spectroscopy.”

The same is true for a two-photon transition and two-photon spectroscopy: the, e.g., 1S-2S transition is not “Doppler-free” and can be driven in a Doppler/velocity-sensitive way by, e.g., using only a single beam. It is only through the arrangement of two counter-propagating beams that the Doppler shift is suppressed in Doppler-free two-photon spectroscopy, just like for our technique of Doppler-free one-photon spectroscopy. Importantly, misalignments will lead to residual first-order Doppler shifts in both cases (one may take the above case of a single beam as the limiting case of misalignment of two beams).

However, as the referee correctly points out, one important distinction is that Doppler broadening is suppressed in Doppler-free two-photon spectroscopy, while it is still present in Doppler-free one-photon spectroscopy. To clarify this, we now write (in Section 2.2): “In the ideal case of laser beams with identical wavefront curvature and power, this excitation scheme produces a line shape whose center-of-mass is free of first-order Doppler shifts (but not necessarily free of Doppler broadening), as the interaction with the respective beams results in Doppler shifts of equal magnitude but opposite sign.”

We believe that calling our technique “Doppler-free one-photon spectroscopy” is both appropriate and helpful, because it succinctly summarizes that the technique removes the (first-order) Doppler shift from the measured transition frequency, just as is the case for Doppler-free two-photon spectroscopy. It is this removal of the Doppler shift, and not the suppression of Doppler broadening, that is of highest relevance for a transition frequency measurement.

11: In the caption to Fig. 2 (c) we read, “The FWHM (full width at half maximum) linewidth Γ reduces for slower velocity groups as Doppler broadening reduces”.

-> The experimental data shows Doppler broadening when the measurement was said to be “Doppler-free” in Section 1 (see above), perhaps the term “Doppler-free” should not be used as it may confuse the reader.

Please see the response to comment 10 above.

12: In the spectra of Fig. 2(c), the error bars in the plot are too small, making it difficult to see how consistent they are with the simulated lineshape; the goodness-of-fit is vitally important in an experiment that determines the centre-of-gravity of the lineshape to 1 part in 15000 of the

linewidth. It would therefore be extremely useful to plot the error residuals ([Exp. data]-[Simulation]) including the error bars in a separate plot beneath Fig. 2(c).

We emphatically agree that goodness-of-fit is vitally important for this experiment. This is why we have studied the fit residuals with great care, the results of which we show in Extended Data Fig. 1 (previously Fig. A2). Overall, we find good agreement between the fit residuals of fits to the experimental data, on the one hand, and fits to the simulations, on the other hand, showing that our simulations capture the relevant line shape features.

We have added a reference to Extended Data Fig. 1 in the caption of Fig. 2 to make these results more visible. Showing the fit residuals of the single line scan of Fig. 2 would not allow the reader to see the goodness-of-fit relevant to the level of precision demonstrated here. This is simply because we need to average over hundreds of line scans to get to our level of precision and to see the goodness-of-fit on this level. This is what's shown in Extended Data Fig. 1, and we therefore choose not to show the fit residuals of a single line scan in Fig. 2.

13: In Fig. 2(c), the wings of each of the three spectral profiles seem to be defined by only two data points at detuning frequencies of -22 MHz and +22 MHz. A symmetrically equal number of datapoints are positioned at the low-frequency (left) and high-frequency (right) sides of each peak at detunings <10 MHz very close to the peak. In other experiments, such a highly non-equidistant choice of data points might be generally considered a risky data-taking strategy: if we choose the positions of the data points to most efficiently resolve the characteristics of a pre-existing model (in this case the centroid), we are in fact increasing the risk of biasing the outcome to agree with our expectations since this dataset choice arguably reduces the chances of detecting a significant deviation between the experiment and simulated lineshape. From this spectra alone we cannot tell how well the theory reproduces the spectral shape at larger laser detunings; perhaps there might be a small constant offset or other unexpected structure superimposed on the Voigt-like function?

The full range of detunings used in the experiment is actually +/-50 MHz, with two more data points on each side not shown in Fig. 2 (c). To clarify this, we added the full detuning range in the caption and a reference to the Methods, where the line sampling is described in detail. We in fact use the +/-50 MHz data points as constraint on a non-constant offset (a constant offset has no effect), as detailed in Section S2.6 of the Supplementary Methods and listed in the uncertainty budget ("Signal background").

We would like to point out that we have studied the sampling bias from our line sampling in detail, as described in Section S2.5 in the Supplementary Methods, with the resulting uncertainty listed in the uncertainty budget. Overall, we find that by far the largest effect is the offset between the center of the detunings - which is subject to laser drifts - and the actual resonance - which is unknown a priori.

To address the question of how to best sample the line: from purely a line sampling perspective, it would be of course best to sample with an even larger number of points. However, the total measurement time is inherently limited, and trading, e.g., a larger number of points for less line scans would reduce the ability to characterize drifts, and so on - that is, there is always a trade-off involved. One choice, as mentioned by the referee, is equidistant sampling, but this choice will tend to either mean not sampling the far wings or poor sampling of the resonance itself. It is also not immediately clear to us that equidistant sampling necessarily leads to the lowest bias.

Our line sampling is a compromise between placing many points on the resonance itself (i.e., within a few linewidths), while also sampling the wings of the resonance. The reasoning is that line shape distortions will have the strongest effect within the resonance, so we should place a stronger weight there (one may think of the weight of a data point in the context of fitting as

roughly given by the derivative of the line shape at that point; note that our line shape is still described by a Voigt line shape within a few percent, so any corrections do not change the derivative substantially). This is also the case for our known systematic line shape distortion (quantum interference and light force shift). In particular, QI both distorts the resonance and leads to a sloping background, with the resulting fit residuals actually larger for larger detunings (see Extended Data Fig. 1 (previously Fig. A2)) - nevertheless, the resulting QI shift is mostly caused by the distortion of the resonance (for details, see Udem et al., *Annalen der Physik* 531, 1900044 (2019)). Our reasoning is that, overall, we place equal weight on potential line shifts caused by distortions within the resonance (larger shift for given distortion and larger point density) and outside the resonance (lower shift for given distortion and lower point density). We then studied the fit residuals and found them to be well-explained by our known systematics (see Extended Data Fig. 1).

14: *"The excited 6P levels rapidly decay, directly or through cascades, to the 1S and 2S manifolds, resulting in a $\Gamma = 3.89$ MHz natural transition linewidth."*

 *I summed the single-photon transition rates of the relativistic calculations tabulated for example in W.L. Wiese and J.R. Fuhr *J. Phys. Chem. Ref. Data* 38, 565-726 (2009), and got a number like 3.90 MHz instead of 3.89 MHz though I am not confident about this number.*

First, many thanks to the referee for thoroughly cross-checking the natural linewidth; we deeply appreciate this in-depth review of our results.

The observed discrepancy is caused by our omission of the reduced-mass correction (a scaling by $(m_p+m_e)/m_p \sim 1.0005$) in the dipole moments in our simulations. Including this correction, we find the same linewidth as the referee (3.90 MHz). We have investigated how the omission of this correction affects our frequency corrections and detail the procedure and resulting changes below.

Overall, we find the effect of the correction to be negligible. None of our main results are affected, including the determined transition frequencies, the proton radius, and Rydberg constant. This is because the resulting correction is at most 0.2 % of our frequency corrections, which are at most of the order of 1 kHz. In the main text, the only visible change is a 10 Hz shift in the number given for the Doppler shift correction for a subset of the data. We list all resulting changes, however negligible, below.

In detail, we have compared our non-relativistic (NR) calculations of the transition rates against the relativistic calculations of Wiese and Fuhr and find a relative deviation of $1e-3$. This is caused by our omission of the reduced-mass correction in the dipole moments, which leads to a relative correction of $((m_p+m_e)/m_p)^2 \sim 1.001$ in the transition rates. After accounting for the reduced mass in the dipole moments, we find agreement in the transition rates with Wiese and Fuhr to $\alpha^2 \sim 5e-5$, as expected for NR vs relativistic calculations.

In particular, the 6P natural linewidth is found to be 3.899 MHz from both NR and relativistic calculations. We therefore have corrected the value given in the text to 3.90 MHz. For reference, we now also give all relevant transition rates to 4 significant digits (on which NR and relativistic calculations agree) in Section S1.2 in the Supplementary Methods.

We note that this only affects the calculation of dipole moments and derived quantities, but not transition frequencies, which are calculated according to our Eq. (1) and include reduced-mass corrections.

The light force shift scales as the Rabi frequency squared (see our Supplementary Methods), and therefore the reduced-mass correction in the dipole moments leads to a correction by $((m_p+m_e)/m_p)^2 = 1.00109$. We have adjusted our LFS frequency corrections by scaling

with this factor (see below). We have also repeated some OBE simulations with the corrected dipole moments, which confirmed this scaling.

The quantum interference shift scales, in leading order (see first term in Eq. (11) of Udem et al., *Annalen der Physik* 531, 1900044 (2019)), with the ratio of the squared linewidth to the frequency separation of the interfering levels (it also depends on a ratio of dipole moments, but the reduced-mass corrections therein consequently cancel). Because we include the reduced-mass correction in the frequencies, accounting for the reduced-mass correction in the dipole moments leads to a correction by $((m_p+m_e)/m_p)^4 = 1.00218$. We have adjusted our QI frequency corrections by scaling with this factor (see below). We have also repeated some OBE simulations with the corrected dipole moments, which confirmed this scaling.

The leading-order quadratic dc-Stark shift scales with the squared dipole moment (and the inverse frequency separation, but here the reduced-mass correction has been taken into account). Therefore, the reduced-mass correction in the dipole moments leads to a correction by $((m_p+m_e)/m_p)^2 = 1.00109$. We have adjusted our dc-Stark shift frequency corrections and quadratic coefficients by scaling with this factor (see below).

Likewise, we have included the reduced-mass correction in the dipole moments in our calculation of the blackbody-radiation-induced shift. However, none of our results are affected by this correction.

Including the reduced-mass correction in the dipole moments leads to the following changes:

- Main text Fig. 2:
 - Panel (a): Changed 6P natural linewidth from 3.89 MHz to 3.90 MHz.
 - Panels (d) and (e): Changes in frequencies by 1 Hz, not visible with the naked eye.
- Main text Fig. 3:
 - Panels (a) and (b) have been updated with scaled simulation results, but the resulting changes are not visible with the naked eye.
 - Panel (d) has likewise been updated; the transition frequency difference for $\alpha_0 = 0$ mrad changed from -2.06(99) kHz to -2.05(98) kHz (the uncertainty reduction is an artifact of rounding the contributing uncertainties, with the LFS correction uncertainty here subtracted in quadrature).
- Main text Fig. 4: Changes in transition frequencies by at most 7 Hz and in Doppler slopes by at most 0.03 Hz/(m/s).
- Main text Section 3, Light force shift:
 - The Doppler shift correction included in Eq. (3) changed by -10 Hz from -1.91(1.81) kHz to -1.92(1.81) kHz.
- Extended Data Fig. 3 (previously Fig. A3), determination of stray electric fields: the simulated transition frequencies under an applied electric field shown in panels (c) and (d) have been corrected (not visible with the naked eye). The resulting dc-Stark shift coefficient remains unchanged.
- Methods, Quantum interference shift:
 - The maximum QI shift at $\theta_L = 56.5$ deg (magic angle) for the 2S-6P 1/2 transition changed from -0.86(54) kHz to -0.87(54) kHz.
 - The maximum QI shift for the 6P FS centroid at any polarization angle changed from -0.35 kHz to -0.37 kHz. This change is largely from using a more conservative definition of maximum shift, found to be more appropriate when cross-checking these numbers, not from the reduced-mass correction.
 - The difference in Doppler-free transition frequency between $\theta_L = 146.5$ deg and 56.5 deg for the 2S-6P 1/2 transition changed by -10 Hz from 0.00(1.69) kHz to -0.01(1.69) kHz, and the corrected-for QI shift changed from 3.43(92) kHz to 3.44(92) kHz.

- The difference in Doppler-free transition frequency between $\theta_L = 146.5$ deg and 56.5 deg for the 2S-6P 3/2 transition changed by 10 Hz from 4.07(1.77) kHz to 4.08(1.77) kHz.
- Methods, dc-Stark shift: Unrelated to the reduced-mass correction, the correlation coefficient between the corrections for the dc-Stark shift for the 2S-6P 1/2 and 3/2 transitions given in the Methods section of the main text was found to contain a mistype: it should read -0.30, not 0.30. No other changes in this Methods section.
- Extended Data Table 4 (previously Table A3): The correction for the first-order Doppler shift for the 2S-6P 1/2 transition has changed by +2 Hz. This changes the value given in Table 3 (rounded to 10 Hz) from -0.33 kHz to -0.32 kHz. This is because the QI and LFS corrections are applied to the resonance frequencies before the Doppler extrapolation is performed and therefore changes in the QI and LFS corrections can affect the Doppler shift correction.
- Extended Data Table 5 (previously Table A5): Some of the contributions to the simulation correction uncertainties have changed, by at most 1 Hz, after accounting for the reduced-mass correction in the QI or LFS corrections.
- SM Section S1.4, Modeling of dc-Stark shift:
 - The dc-Stark shift coefficient from second-order perturbation theory for the 2S-6P 1/2 transition and for the static electric field perpendicular to the quantization axis changed from $-1.50 \text{ kHz}/(\text{V}/\text{m})^2$ to $-1.51 \text{ kHz}/(\text{V}/\text{m})^2$.
 - The dc-Stark shift coefficient from simulations for the 2S-6P 3/2 transition, the static electric field perpendicular to the quantization axis, and when observing Lyman-alpha decay changed from $-17.66 \text{ kHz}/(\text{V}/\text{m})^2$ to $-17.68 \text{ kHz}/(\text{V}/\text{m})^2$, which we now give with 3 significant digits as $-17.7 \text{ kHz}/(\text{V}/\text{m})^2$.
 - The dc-Stark shift coefficient from simulations for the 2S-6P 3/2 transition, the static electric field perpendicular to the quantization axis, and when averaging over all Lyman decay changed from $-0.56(74) \text{ kHz}/(\text{V}/\text{m})^2$ to $\beta_{\text{dc,perp}} = -0.56(75) \text{ kHz}/(\text{V}/\text{m})^2$.
 - The dc-Stark shift coefficient from second-order perturbation theory for the 2S-6P 3/2 transition, the static electric field parallel to the quantization axis, and all perturbing levels included changed from $-33.57 \text{ kHz}/(\text{V}/\text{m})^2$ to $-33.60 \text{ kHz}/(\text{V}/\text{m})^2$, which we now give with 3 significant digits as $-33.6 \text{ kHz}/(\text{V}/\text{m})^2$.
 - Because of the change in the relevant $\beta_{\text{dc,perp}}$, the dc-Stark shift coefficient for the 2S-6P 3/2 transition in the stray-field regime and along the detector's z-axis (for $\theta_L = 56.5$ deg) changed from $-0.16(23) \text{ kHz}/(\text{V}/\text{m})^2$ to $\beta_{\text{dc,z}} = -0.16(24) \text{ kHz}/(\text{V}/\text{m})^2$. We have also updated the corresponding correlation coefficients (with now all three given).
 - In addition, unrelated to the reduced-mass correction, the dc-Stark shift coefficient for the 2S-6P 3/2 transition in the bias-field regime and along the detector's y-axis (for $\theta_L = 56.5$ deg) was found to be a mistype: it has been corrected from $-0.52(2) \text{ kHz}/(\text{V}/\text{m})^2$ to $\beta_{\text{dc,y}} = -0.53(3) \text{ kHz}/(\text{V}/\text{m})^2$.

15: The authors state “ $\gamma_{\text{ei}}/\Gamma = 3.9\%$ or 7.9% of decays from the excited level lead back to the initial 2S level for the 2S-6P1/2 or 2S-6P3/2 transitions, respectively.”

-> Perhaps the sentence should not be started by an equation. I suppose by “initial 2S level” you actually mean the $2S_{F=0, m_F=0}$ sublevel?

We added "A fraction of" to the start of the sentence to avoid it starting with an equation. We now explicitly state that the $2S_{12}^{F=0, m_F=0}$ level is our initial level by writing "we alternately probe two dipole-allowed transitions from the initial (metastable) $2S_{12}^{F=0}$

mF=0 level" instead of "[...] from the metastable 2S_{1/2} F=0 mF=0 level". The "metastable" is placed in parenthesis to avoid the reading as (nonsensical) "initially metastable".

16: *"The 6P fine-structure splitting $\Delta\nu_{FS}(6P) \approx 405$ MHz between the two excited levels"*

-> *Four levels are indicated for the 6P manifold in Fig. 2 which may be confusing for some in the general readership. Perhaps the uncertainty of the 405 MHz value can be indicated in this sentence.*

The other levels shown in Fig. 2 (a) are part of hyperfine manifolds not coupled by the laser excitation scheme used here and are shown in gray instead of black as explained in the caption. However, some of these levels (1S F=1 and 2S F=1) are populated by decays. In particular, the 2S F=1 hyperfine manifold is important to show, as this is where the decays to the 2S manifold that do not go to the initial 2S F=0 level lead. The reader might otherwise wonder where the remaining population ends up. For consistency, we also show the non-coupled (by excitation or decay) 6P 1/2 F=1 and 6P 3/2 F=2 manifolds. In addition to those non-coupled levels being color-coded, there are two arrows for the two probed transitions, clearly pointing out which two of the four shown 6P levels are coupled here. Because of this, we think showing the non-coupled levels has less potential for confusion than not showing them.

We give the values and uncertainties of the QED prediction and the measurement of the 6P FS splitting below and in Eq. (6). At the point discussed here, we do not want to concern the reader with distinguishing between these two numbers but would like to give the relevant scale of the splitting, which is why we do not indicate the uncertainty at this point.

17: *"Nevertheless, quantum interference (QI) between excitation–decay paths through either excited fine-structure manifold can lead to substantial distortions of the line shape"*

-> *The non-specialist may be baffled by the adverb "nevertheless". A very brief explanation of the quantum interference effect, its dependency on the natural width and spectral spacings of multiple resonance lines, and why it may potentially affect the resonance lineshape is required here. Also needed are the reasons why the quantum interference is suppressed at the magic angle of 56.5 degrees of the laser polarization angle and by the increased solid angle of the detector, as these details are mentioned later in the main text without an accessible explanation. A read of Section 4 also provides no explanation of what quantum interference might be.*

Thank you for bringing this up, we agree that further explanation of quantum interference is necessary for non-specialists. We've expanded the brief discussion of QI at the end of Section 2.1, which now explains the underlying physical process, the connection of the order of magnitude of the associated shifts with the linewidth and fine-structure splitting, and the dependence of the shift on the detection geometry and laser polarization angle. We've also made some small edits to the relevant sentence in Section 1.2 of the Supplementary Methods.

18: *In Section 2.2, the authors state "...offset angle $|\alpha_0| = 0$ mrad . . . 12 mrad from the orthogonal". Later we read that "We align α_0 close to zero".*

-> *At first read it is hard to interpret whether the indicated range of angles is caused by the divergence of the atomic and laser beams (mentioned in the previous paragraph of the main text), or represent the offset angles that were intentionally scanned to evaluate systematic uncertainties. Later in the paper and in Methods we find that the latter is true.*

We've edited "...with an [...] offset angle..." to "...at an adjustable [...] offset angle..." to clarify that the range given here is not a divergence, but the range of values to which the offset angle is set to in the experiment. Regarding the apparent contradiction with the later statement "We align α_0 close to zero", please note that the complete sentence reads: "We align α_0 close to zero [...], except when characterizing the light force shift (LFS; see below)."

19: "wavefront-retracing"

-> This may be jargon; I could only find a few groups that use this terminology in publications searchable by Google Scholar. It should be briefly explained since it constitutes a vital point in the experimental work. What happens when the wavefront is not "retraced"?

We've replaced "wavefront-retracing laser beams with identical power" with "laser beams with identical wavefront curvature and power" to briefly introduce our scheme while avoiding specific terminology.

The term "wavefront-retracing" is kept in the next sentence, as it is used in our previous publications on our Doppler suppression scheme ([27-28] Wirthl et al. (2021), [26] Beyer et al. (2016)). The cited Wirthl et al. (2021) describes the exact experimental implementation used for the measurement presented here and, in particular, explores the resulting Doppler shifts if the wavefronts are not perfectly retraced.

20: The term "dc Stark shifts" is mentioned several times in the main text without explaining what they are in the main text. A brief sentence explaining why the shift has (mostly) a quadratic dependence to the electric field may aid the reader.

Thank you, we have made changes to the text in two positions:

To clarify to the non-expert that dc-Stark shift are shifts caused by stray electric fields, we have added "caused by such fields" to "A segmented Faraday cage [...] allowing the application of bias fields to characterize stray electric fields and the dc-Stark shift caused by such fields (see Methods)."

To briefly explain the origin of the quadratic nature of the shift (which is not directly relevant in the main text), we added a sentence in the corresponding Methods section: "The quadratic behavior arises because the energy levels contributing to the net shift are well-separated in energy from the perturbed level in either regime."

21: In Section 2.3 we read "distortions from the LFS dominate, for which no equivalent line shape model is known to us"

-> The acronym LFS appears without explanation, though it is defined later in Section 3.

Please note that the acronym LFS is first defined in the preceding Section 2.2. We therefore have not made any changes to address this comment.

22: Perhaps some of the highly interesting but detailed explanations of Section 3 can be moved to either Methods or Supplementary Materials, to make space for other vital explanations that would make the paper more accessible to the general reader.

We will keep this in mind in case we need to shorten the main text.

23: "This precision is comparable to the test of the Standard Model prediction of the electron magnetic moment, which is currently limited to 0.7 ppt by discrepant measurements of a [11]."

-> *This may be a matter of taste, but a measurement of the $g-2$ anomalous magnetic moment of a single lepton, and a measurement of a two-body lepton-hadron bound system, are probing related but quite different things and so a comparison of the relative precision may not be so informative. Of greater importance is the fact that both the electron $g-2$ and hydrogen experiments and theory reach the unprecedented precision of three-photon corrections.*

As we argue in our response to the referee's first comment, our experiment is part of a class of experiments probing the Standard Model, with the $g-2$ measurement being a prominent member of that class. We therefore think it appropriate to compare the experiments here, and leave the sentence in the manuscript.

We agree that the fact that both experiments reach the level of three-photon corrections is important, which is why we highlight it in the main text shortly thereafter.

24: *In Methods we read, "which we attribute to fluctuations in the atomic flux (of metastable 2S atoms)"*

-> *What is the reason for the parenthesis?*

We refer to the flux of metastable 2S atoms as "atomic flux" throughout this Section. The parenthesis defines that "atomic flux" is here not, e.g., the flux of atoms out of the nozzle, but the flux of atoms that have been excited to the 2S level (which depends both on the flux of ground-state atoms out of the nozzle and the laser excitation to the 2S level). We prefer to leave the current wording.

25: *"more refined spatial detection efficiency simulations"*

-> *5 modifiers/adjectives*

We have replaced this unwieldy expression with "more refined simulations of the spatial detection efficiency".

26: *"Importantly, while Eq. (21) is found to approximately hold in either regime, the coefficient β_{dc} may differ."*

-> *A brief sentence explaining why the coefficient is allowed to differ in the two experimental regimes may be useful. What happens if we insist that the coefficient be the same in both regimes, as may be naively expected?*

We have added the following sentences in the corresponding Methods section (see also response to comment #20): "The quadratic behavior arises because the energy levels contributing to the net shift are well-separated in energy from the perturbed level in either of our regimes. However, the shift of the involved levels between the regimes can lead to substantially different energy separations and thereby different values of β_{dc} ."

The coefficient differs significantly between the two regimes only for the 2S-6P_{3/2} transition when the stray electric field is parallel to the quantization axis. Here, we find $\beta_{dc} = -0.55(3)$ kHz/(V/m)² and 0.56(75) kHz/(V/m)² in the bias-field and stray-fields regimes, respectively. We use the latter to correct the dc-Stark shift from stray fields.

In the hypothetical scenario where we hadn't calculated the coefficient in the stray-field regime but had only measured it in the bias-field regime, we would be (erroneously) using the bias-field value. Assuming, as an upper limit, a stray electric field of 0.4 V/m parallel to the quantization

axis, this would result in a correction of +90(5) Hz instead of -90(120) Hz, i.e., a shift of +180 Hz. This corresponds to 0.28 sigma for the 2S-6P3/2 transition frequency and would be insignificant.

Response to referee 3

Referee comments are italicized; our (the authors') responses are upright and in blue.

The manuscript under review addresses the measurement of the 2S–6P transition in atomic hydrogen. Spectroscopic measurements in light atoms provide the most precise low-energy tests of the Standard Model of fundamental interactions, with the hydrogen atom being the most prominent case. Together with the extremely accurate 1S–2S transition measurement, one additional high-precision transition is required in order to extract both the Rydberg constant and the proton charge radius, on which binding energies depend. The authors accomplish exactly this: their measurement of the 2S–6P transition is so precise that the resulting value of the proton charge radius surpasses the accuracy of all previous determinations from atomic hydrogen. Moreover, it is in excellent agreement with the value obtained from muonic hydrogen. This is a very significant result, given the longstanding discrepancies in proton charge radius determinations from atomic hydrogen. The results presented in this work are therefore of broad importance to the physics community and, in my opinion, merit publication in Nature.

Authors are experienced and respected experimentalists with a long record of measurements on hydrogenic systems (both ordinary and muonic). The present work represents a continuation of their previous efforts in the field (in this case their 2S-4P measurement, which is cited in the manuscript). The manuscript consists of the main section, a methodology section, and the Supplemental Material. The main section provides a clear description of the principle of experiment. The most important contributions affecting the determination of resonance frequencies (Doppler effect, quantum interference and light force shift corrections) are briefly discussed, with detailed treatments deferred to the Methods section and the Supplemental Material. The results obtained for the two transition frequencies into 6P fine structure levels are highly impressive and constitute exceptionally accurate test of the Standard Model predictions. The conclusions drawn in the manuscript are valid and well supported.

The methodology section presents specific details of the experiment, including statistics, modeling and data analysis. The presentation is thorough, transparent and the discussion of error bars is sound and well justified. All data (including the tables and figures in Appendix) are presented clearly. The Supplemental Material provides further details of the models used in the measurement and the interpretation of results, which I find satisfactory. It also includes a discussion of less significant corrections that are included in table A3. Overall, the manuscript is clearly written and accessible.

To summarize my report: I recommend the publication of this manuscript in Nature. The transition frequencies reported in this work allow for extremely accurate test of the Standard Model (on level of ppt) and yield the most accurate determination of the proton charge radius from atomic hydrogen to date. This result is in excellent agreement with the corresponding value from muonic hydrogen and contributes to resolving the proton radius puzzle. For these reasons, I believe this work fully meets the criteria for publication in this journal.

Minor correction: in Supplemental Material, on line 233: "which" should probably be "with"

Thank you, this has been corrected.